# Exploring non-equilibrium processes and spatio-temporal scaling laws in heated egg yolk using coherent X-rays

Nimmi Das Anthuparambil [1,2] ✉, Anita Girelli [3], Sonja Timmermann [2], Marvin Kowalski [2], Mohammad Sayed Akhundzadeh[2], Sebastian Retzbach [3], Maximilian D. Senft [3], Michelle Dargasz [2], Dennis Gutmüller[3], Anusha Hiremath [3], Marc Moron [4], Özgül Öztürk [2], Hanna-Friederike Poggemann [3], Anastasia Ragulskaya [3], Nafisa Begam [3], Amir Tosson [2], Michael Paulus [4], Fabian Westermeier [1], Fajun Zhang [3], Michael Sprung [1], Frank Schreiber [3] & Christian Gutt [2] ✉

The soft-grainy microstructure of cooked egg yolk is the result of a series of out-of-equilibrium processes of its protein-lipid contents; however, it is unclear how egg yolk constituents contribute to these processes to create the desired microstructure. By employing X-ray photon correlation spectroscopy, we investigate the functional contribution of egg yolk constituents: proteins, low-density lipoproteins (LDLs), and yolk-granules to the development of grainy-gel microstructure and microscopic dynamics during cooking. We find that the viscosity of the heated egg yolk is solely determined by the degree of protein gelation, whereas the grainy-gel microstructure is controlled by the extent of LDL aggregation. Overall, protein denaturation-aggregation-gelation and LDL-aggregation follows Arrhenius-type time-temperature superposition (TTS), indicating an identical mechanism with a temperature-dependent reaction rate. However, above 75 °C TTS breaks down and temperature-independent gelation dynamics is observed, demonstrating that the temperature can no longer accelerate certain non-equilibrium processes above a threshold value.

Egg yolk is widely utilised as a culinary component due to its high nutritious value[1] and excellent emulsifying and gelling abilities[2–5]. When heated, it undergoes a solution-to-gel transition and although this looks very simple from a cooking point of view, the final gel microstructure is the result of a series of out-of-equilibrium processes of its protein and lipid contents. Typical processes involved during heating are fusion, denaturation, aggregation, and gelation which are coupled by a hierarchy of length, time and energy scales[6,7]. Rheology[3,4,8–10] and microscopy[9,10] have been used to study the viscoelastic properties and structure of yolk gel, providing an average picture of the final gel and its elastic properties. However, the temporal evolution of the structure of its main constituents such as the yolk-plasma proteins, low-density lipoproteins (LDLs), and yolk-granules during heating including the formation of the grainy nanoscale structure is not well understood. Similarly, the time scales and dynamics of the nano- and microscopic processes that contribute to the macroscopic viscosity and the microstructure of the egg yolk gel are also still unclear despite the omnipresence of egg

[1]Deutsches Elektronen-Synchrotron DESY, Notkestr. 85, 22607 Hamburg, Germany. [2]Department Physik, Universität Siegen, 57072 Siegen, Germany. [3]Institut für Angewandte Physik, Universität Tübingen, 72076 Tübingen, Germany. [4]Fakultät Physik/DELTA, Technische Universität Dortmund, 44221 Dortmund, Germany. ✉e-mail: nimmi.das.anthuparambil@desy.de; christian.gutt@uni-siegen.de

yolk in our kitchens, in the food industry[2,5,11], and also in biotechnology[12,13].

Apart from its food, biological, and therapeutic value[2,14], the diversity of proteins and high concentration of LDLs make egg yolk also an ideal candidate for studying the physics behind biologically relevant non-equilibrium processes. Most often denaturation and aggregation of proteins and LDLs are undesirable in biological systems, especially in relation to the pathogenesis of several human diseases such as Alzheimer's disease[15], Parkinson's disease[16], atherosclerosis[17], and others[18]. Furthermore, the instability and aggregation of protein drugs, caused by deviations from optimal conditions of temperature, pH, ionic strength, etc. are one of the main challenges the pharmaceutical industry faces[19]. Thus the nanoscopic non-equilibrium processes involved in protein aggregation need to be understood on the relevant time and length scales which would also aid other applications[5,20,21] as well. However, often the complexity of a multi-component protein sample and the need to experimentally monitor simultaneously a large window of length and time scales renders experimental insights difficult.

Here, we employ low-dose X-ray photon correlation spectroscopy (XPCS)[22–26] in ultra-small angle X-ray scattering (USAXS) geometry and investigate the contribution of the yolk constituents plasma proteins, LDLs, and yolk-granules, to the formation of the grainy-gel microstructure under heat induction. We find that at temperatures below 75 °C only the proteins of the yolk-plasma undergo gelation and form a protein gel network, whereas at higher temperatures (>75 °C), fusion and aggregation of LDLs along with protein denaturation result in the formation of a grainy-gel-microstructure. The relaxation times of the yolk are determined by the extent of protein gelation and are unaffected by the nanoscale structural changes that occur during LDL aggregation.

We observe that the temporal evolution of the structure of the protein gel network, the aggregation of LDLs and the microscopic relaxation rates each follow particular scaling laws, which collapse onto master curves when scaling the waiting times during cooking with their respective characteristic time scales. The scaling parameters largely follow an Arrhenius temperature behaviour and thus the overall kinetic and dynamical evolution governing protein denaturation-aggregation-gelation and LDL aggregation follows Arrhenius-type time-temperature superposition (TTS)[27]. This implies identical mechanisms with temperature-dependent reaction rates. However, above 75 °C, the TTS behaviour breaks down and instead a temperature-independent gelation dynamics is observed. This indicates that, remarkably, thermal energy can no longer accelerate certain non-equilibrium processes above a temperature threshold pointing toward the presence of intrinsic time scales.

Finally, we construct from the data a generic time-temperature phase diagram for the out-of-equilibrium processes during egg yolk gelation. The diagram illustrates the coupling of the nanoscale processes that give rise to the gel structure in a wide range of time-temperature combinations. We expect that these results are relevant beyond egg yolk and that similar scaling laws will accompany a large range of denaturation phenomena and nanoscale structure formation in dense protein systems.

## Results

The heat-induced gelation of egg yolk is the result of a series of out-of-equilibrium processes coupled by a hierarchy of length and time scales. While the microstructure of the system changes from a protein-lipid solution to a soft grainy-gel network via protein gelation and LDL aggregation, the viscosity is expected to show an exponential increase at this transition[8]. In the context of food science, this indicates the onset of changes in texture and correlates with the mouth feel of food. In general, these processes are also relevant for the fundamental

understanding of nano- to micro-scale structure formation in concentrated protein/lipid systems. This implies that a complete understanding of these complex non-equilibrium processes necessitates a simultaneous understanding of changes in the structure and dynamics of its components[28,29]. XPCS is a potential solution to accomplish this goal.

Figure 1a shows a sketch of the experimental setup used in this study. A coherent X-ray beam is scattered by the sample contained in a capillary in transmission geometry. The scattered X-rays are recorded by a fast area detector at a distance of 21.2 m from the sample (see Methods for further details). This configuration provides access to scattering wave vectors, $q$, in a range from ≈0.005 nm$^{-1}$ to ≈0.2 nm$^{-1}$. While the temporal evolution of the azimuthally integrated scattering intensity profiles (Fig. 1b) provides information about structural changes, the intensity autocorrelations (Fig. 1c), extracted by correlating the temporal scattering intensity fluctuations at a specific $q$ value, provide information about the sample dynamics. The complexity of the egg yolk sample is depicted in Fig. 1a. The egg yolk contains micron-sized non-soluble aggregates called granules which are dispersed in a clear yellow fluid called egg yolk-plasma that contains LDL and water-soluble proteins (livetins)[3]. More details can be found in Supplementary Fig. 1.

Employing the experimental setup indicated in Fig. 1, we investigated the temporal evolution of structure and microscopic dynamics associated with the heat-induced non-equilibrium processes in hen egg yolk at temperatures in the range of 63 °C to 100 °C. The samples were heated at a rate of 150 °C/min to a final temperature, $T$ and hold at this temperature upon reaching it, while X-ray data series were collected in parallel. Throughout the text, two time scales are used, isothermal waiting time $t_w$ and absolute waiting time $t'_w$, which are related via $t_w = t'_w - t_{heating}$, where $t_{heating}$ is the time taken to reach the final $T$, estimated from temperature calibration curves of Linkam heating stage (see Supplementary Fig. 3). From the time-resolved azimuthally integrated scattering intensity, $I(q)$, we identify two types of structural changes in the system depending on the temperature. This represents two separate non-equilibrium processes that occur at low (63 °C ≤ $T$ < 75 °C) and high (75 °C ≤ $T$ ≤ 100 °C) temperatures, as described in the following sections.

### Protein denaturation at low temperatures 63 °C ≤ $T$ < 75 °C

We first investigate the structural changes observed in the low-temperature regime 63 °C ≤ $T$ < 75 °C. Figure 2a depicts the temporal evolution of representative scattering profiles as a function of $q$ for increasing isothermal waiting times at a sample temperature of $T = 68$ °C ($I(q)$ of other temperatures are provided in Supplementary Fig. 10). Two distinct trends are observed in the low-$q$ (<0.02 nm$^{-1}$) and the high-$q$ (0.02–0.17 nm$^{-1}$) regime. While $I(q)$ in the low-$q$ regime remains unchanged, it increases as a function of isothermal waiting time, $t_w$ in the high-$q$ regime. As depicted in Fig. 2a, $I(q)$ at low $q$ follows a power-law dependence ($I(q) \propto q^{-4}$), indicating Porod scattering[30] contribution of egg yolk constituents. It has to be noted that the radius of yolk-granules is $R \approx 1 \mu m$[31–33] and Porod scattering is expected for $q > \pi/R \approx 0.003$ nm$^{-1}$ [34]. Hence the scattering contribution at low $q$ originates from the surface scattering of micron-sized yolk-granules. This is further confirmed by comparing the $I(q)$ of yolk, yolk-granules and yolk-plasma as depicted in Supplementary Fig. 9.

The almost constant scattering profiles in the low-$q$ regime reflect the thermal stability of granules at temperatures below 72 °C[35, 36], which is attributed to the large number of phosphocalcic bridges that link high-density lipoproteins (HDLs) and phosvitin inside yolk-granules. The phosphocalcic bridges are responsible for the compact and poorly hydrated structure of yolk-granules, which effectively protects granules from heat-driven aggregation[35–37].

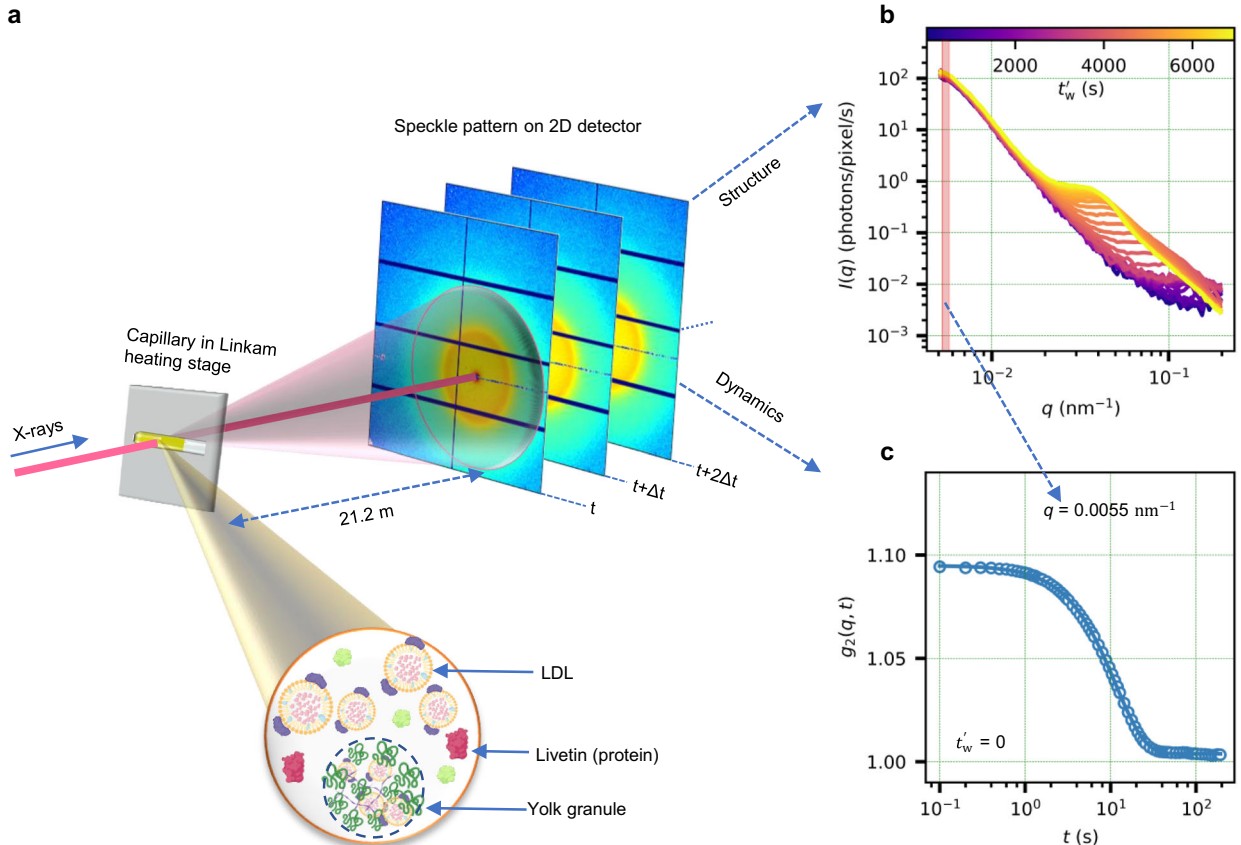

**Fig. 1 | Schematic illustrating the XPCS measurement. a** Coherent X-rays from the synchrotron source are scattered by the sample in a capillary, which is mounted inside a Linkam heating stage. The speckle patterns are collected using a two-dimensional detector at a distance of 21.2 m from the sample. A schematic of different components of the hen egg yolk is shown at the bottom. Egg yolk is an assembly of a variety of proteins (livetins), LDLs, and yolk-granules[3]. The LDLs, yolk-granules, and livetins constitute ≈66%, ≈22%, and ≈10% of yolk dry matter respectively[2,3,86]. LDLs are spherical core-shell molecules (average diameter ≈30 nm) having a lipid core (triglycerides, cholesterol esters, and free cholesterol) surrounded by a shell of phospholipids, cholesterol, and apolipoprotein[2]. The egg yolk-granules are circular complexes (diameter ≈0.3−2 μm[31–33]) made of LDLs, high-density lipoproteins (HDLs), and a protein called phosvitin[36,87]. Parts of this schematic were created using Biorender. **b** Time-resolved azimuthally integrated scattering intensities provide information about structural changes in the system. The absolute waiting time from the beginning of the experiment is denoted by $t'_\mathrm{w}$. **c** The sample dynamics is extracted from the autocorrelation of the scattering intensity at a specific $q$ value.

To quantify the observed increase in intensity in the high-$q$ regime, we calculate the scattering invariant, $Q$,[25,38,39] using

$$Q = \int_{q_1}^{q_2} q^2\, I(q)\, \mathrm{d}q. \tag{1}$$

The lower and upper limit of integration ($q_1 = 0.02\,\mathrm{nm}^{-1}$ and $q_2 = 0.17\,\mathrm{nm}^{-1}$) are indicated by vertical arrows in Fig. 2a. The extracted values of $Q$ normalised with respect to the initial value $Q_0$ ($Q(t_\mathrm{w} = 0)$ before heating, see Supplementary Fig. 11) are shown in Fig. 2b. Clearly, $Q$ increases with $t_\mathrm{w}$ at all temperatures, and the rate of increase in $Q$ is temperature-dependent. In addition, for a given temperature, the slope of the curve increases with $t_\mathrm{w}$. In order to quantify these effects, we model the curves using a power-law ($Q/Q_0 \sim t_\mathrm{w}^\alpha$) at low and high $t_\mathrm{w}$. Interestingly, for all $T < 75\,°\mathrm{C}$, we find two regimes where $Q/Q_0$ follows $\sim t_\mathrm{w}^{0.04}$ and $\sim t_\mathrm{w}^{0.2}$ at low and high $t_\mathrm{w}$, respectively. This suggests a structural transition, and the transition time, $t^*$, is estimated by the intersection of the two power-law fits. The resulting $t^*$ for $T = 65\,°\mathrm{C}$ is shown in Fig. 2b, and the same procedure for other temperatures is given in Supplementary Fig. 12.

The egg yolk proteins, especially $\gamma$-livetin, are known to denature at $T > 60\,°\mathrm{C}$[40]. Hence, the initial slow increase in $Q$ is indicative of heat-induced protein denaturation[40–42]. Following the literature[41–45] on protein denaturation and gelation, we speculate that the quick

structural changes observed after $t^*$ indicate the association of lower-order protein aggregates that results in a three-dimensional gel network. In addition, by normalising the curves in Fig. 2b with respect to $t^*$, we obtain a master curve (Fig. 2c) revealing the TTS relationship.

The $t^*$ follows a linear temperature dependence in a semi-log plot as depicted in the inset of Fig. 2c. The importance of $t^*$ becomes clearer when we compare the dynamical information with the underlying structural evolution in the later sections.

### LDL aggregation at high temperatures $75\,°\mathrm{C} \leq T \leq 100\,°\mathrm{C}$

It is well known that distinct egg yolk constituents have varying thermal stabilities[35,36], and as a result, they undergo thermal denaturation in different temperature ranges. Here, we aim to explore the heat-induced aggregation of LDLs that occur at high temperatures ($T \geq 75\,°\mathrm{C}$). Figure 3a shows the temporal evolution of the scattering profiles as a function of $q$ at $T = 85\,°\mathrm{C}$ ($I(q)$ profiles of other temperatures are provided in Supplementary Fig. 14). The increase in $I(q)$ in the high-$q$ regime, as already observed at low temperatures, occurs at high temperatures as well (the increase in $I(q)$ is more clear for yolk-plasma samples as depicted in Supplementary Fig. 18b), but in a much shorter time window, which is difficult to resolve in our experiments. Interestingly, we observe the emergence of a peak at high $q$ at early waiting times, and its position ($q_\mathrm{peak}$) shifts to smaller $q$ values with increasing $t_\mathrm{w}$. A careful evaluation of the temporal evolution of $I(q)$ in small angle

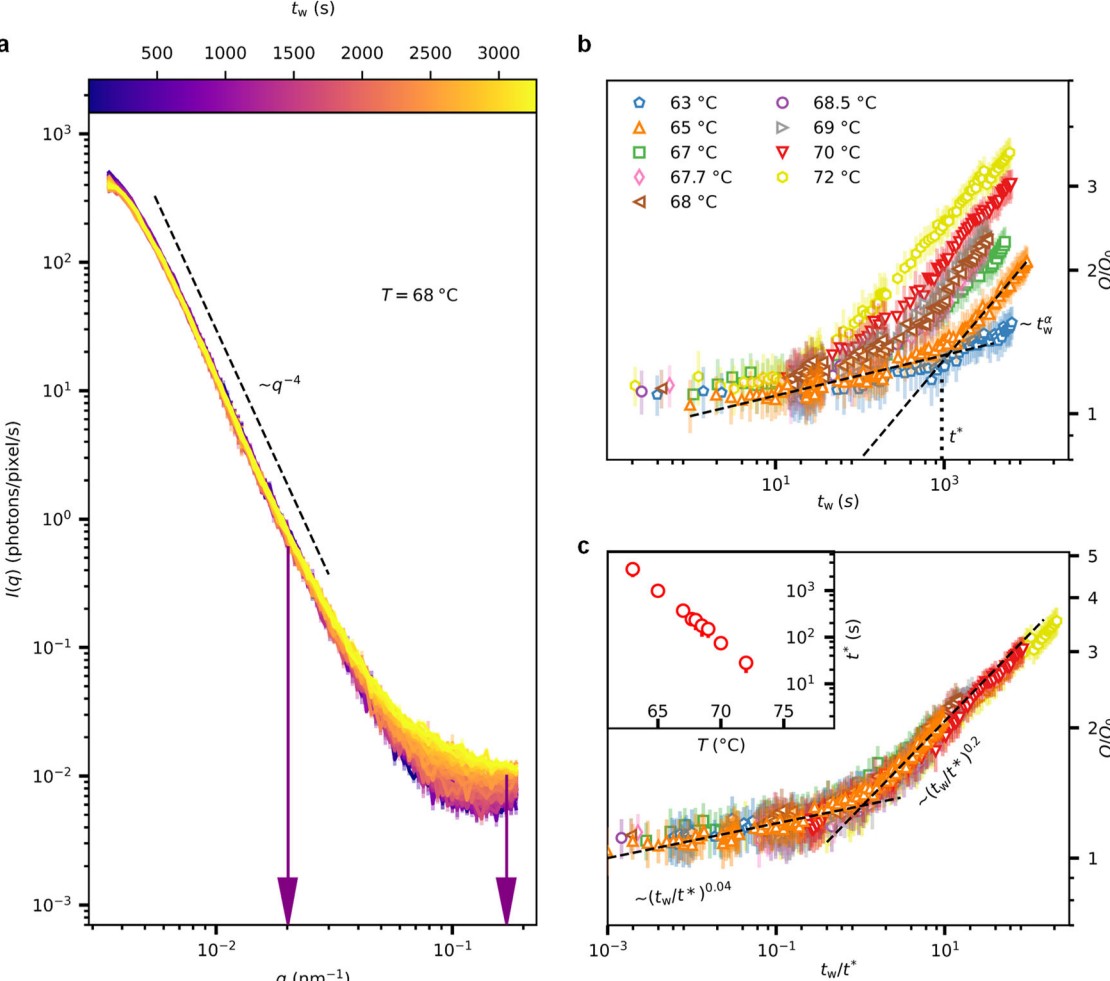

**Fig. 2 | Kinetics of protein denaturation and gelation at low-temperature.**
**a** Scattering profiles collected at a temperature of $T = 68$ °C as a function of iso-thermal waiting time $t_w$ as indicated by the colour bar. The error bars indicate the standard deviation. The dashed line represents $-q^{-4}$. **b** Normalised scattering invariant (using Eq. (1)) estimated from the scattering intensities in the $q$-range 0.02–0.17 nm$^{-1}$ for $T < 75$ °C, as a function of $t_w$. The $q$-range used for the calculation of the scattering invariant is indicated by the vertical arrows in **a**. The normalisation is performed with respect to the initial $Q$ value at $t_w = 0$ ($Q_0$). This normalisation helps to nullify the variations in the scattering intensity of egg yolk samples due to the inherent heterogeneity of the sample volume. The error in $Q$ is estimated by considering the error in $I(q)$. The error bars in $Q/Q_0$ represent the standard

deviation estimated via error propagation. The black dashed lines in **b** represent power-law fits for low $t_w$ and high $t_w$ on the 65 °C data. The functional form of the power-law is $\sim t_w^\alpha$, where $\alpha$ is the power-law exponent. The $\alpha$ values for all temperatures are provided in Supplementary Fig. 13. The intersection point of power-law fits defines $t^*$ for a specific $T$. **c** $Q/Q_0$ as a function of $t_w/t^*$ results in a master curve. The dashed lines in the low $t_w$ and high $t_w$ regime indicate $\sim (t_w/t^*)^{0.04}$ and $\sim (t_w/t^*)^{0.2}$ respectively. The colour code is the same as in **b**. The temperature dependence of $t^*$ is depicted in the inset of **c**. The error bar in $t^*$ is estimated using the error in the fit parameters of power-law fits at low-$q$ and high-$q$ regime. Source data are provided as a Source Data file.

X-ray scattering (SAXS) geometry ($q$-range of ≈0.02–1 nm$^{-1}$) of pure egg yolk-plasma (Supplementary Fig. 18), shows the gradual shifting of the LDL structure factor peak at $q \approx 0.22$ nm$^{-1}$ to lower $q$ values. This indeed confirms that we are probing the fusion and aggregation of LDLs. The width of the aggregate peak and the absence of higher-order structure factor oscillations are indicative of a high degree of dispersity of the aggregates (Supplementary Fig. 18). However, since the LDL size is distributed in the range of 17–60 nm$^3$, the observed dispersity in the aggregates is to be expected. Moreover, the grainy microstructure of egg yolk and dispersity is indeed apparent in the scanning electron microscopy (SEM) images (Supplementary Fig. 34) of heated egg yolk. In addition, a small increase in $I(q)$ at low-$q$ (<0.01 nm$^{-1}$) is indicative of the structural deformation of the yolk granules, which was absent at low temperatures (≤72 °C).

To quantify the kinetics of the LDL aggregation, we also extract the temporal evolution of the correlation length, $\zeta = 2\pi/q_{peak}$ (see Supplementary Note 7). The correlation length extracted from the $I(q)$

of yolk-plasma coincides exactly with that of the full yolk (Supplementary Fig. 17), confirming that LDL aggregation is not affected by the presence of yolk-granules in the surrounding medium. Remarkably, the overall evolution of correlation lengths from XPCS and aggregate sizes from SEM as a function of temperature are in good agreement as depicted in Supplementary Fig. 34d.

A more detailed inspection of individual $\zeta$-curves at different temperatures reveals two distinct growth regimes, as shown in Fig. 3b. Initially, an exponential growth (stage-I) is observed, which is typical of reaction-limited aggregation (RLA) kinetics, in which only a small fraction of particle collisions lead to the formation of an aggregate[46]. The RLA-type aggregation mechanism is reported in the aggregation kinetics of human-LDLs[47] and other colloidal systems[48,49]. As proposed in the literature on human-LDL aggregation[50], heat-induced denaturation leads to the fusion of LDLs, during which the contents of neighbouring LDLs (two or more) merge to create an enlarged particle resulting in an enhanced dispersity. After the fusion, ruptured and

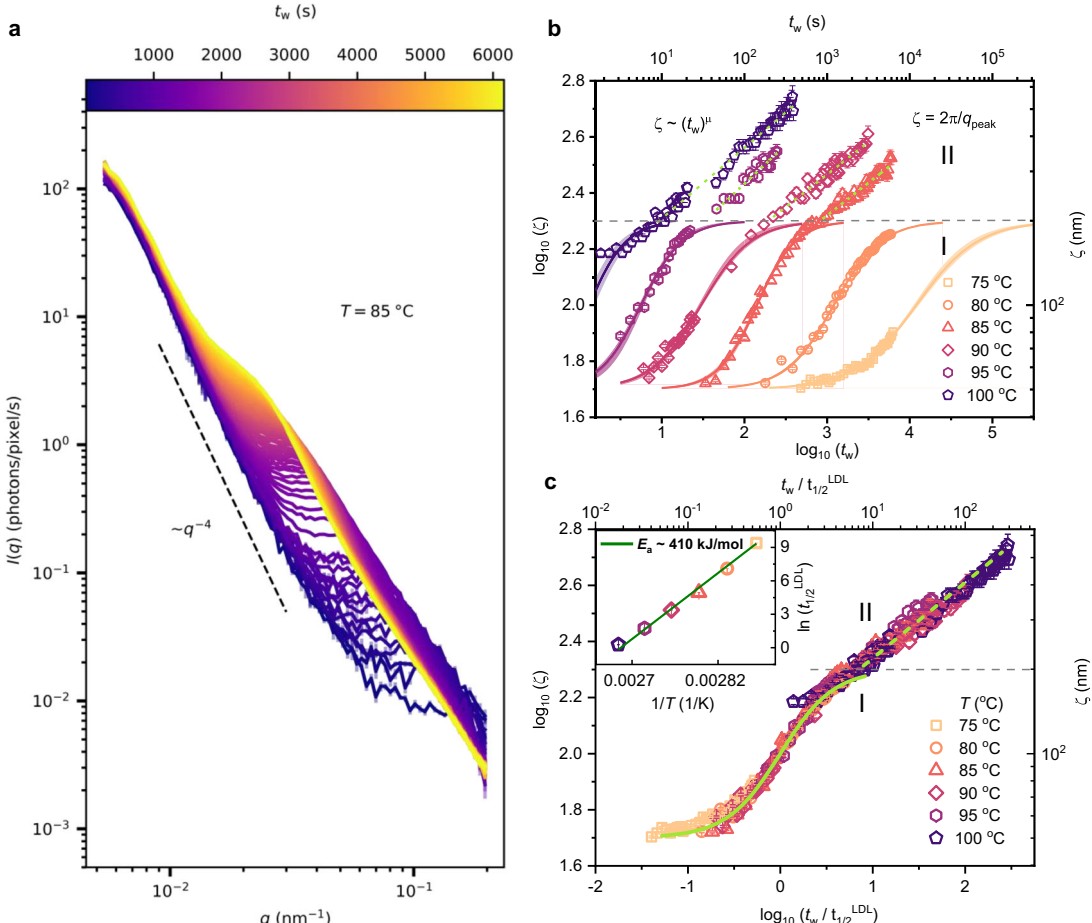

**Fig. 3 | Kinetics of LDL aggregation at high temperatures. a** $I(q)$ profiles collected at a sample temperature of $T = 85\,°C$ as a function of isothermal waiting time, $t_w$ as indicated by the colour bar. The error bars indicate the standard deviation. The dashed line represents $-q^{-4}$. **b** The correlation length ($\zeta = \frac{2\pi}{q_{peak}}$) estimated from the scattering profiles as a function of $t_w$ for sample temperatures $T \geq 75\,°C$. The error bars are obtained from the error in $q_{peak}$ estimation (see Supplementary Note 7). The kinetics of LDL aggregation shows two growth regimes: stage-I - exponential growth phase, and stage-II - power-law growth phase. The solid lines are sigmoidal fits of $\log_{10}(\zeta)$ vs $\log_{10}(t_w)$ using Eq. (5). The light-coloured shaded area around the solid sigmoidal fit lines represents the 95% confidence band. The green dashed lines in stage-II are power-law fits ($\zeta \sim t_w^\mu$). **c** The correlation length as a function of normalised $t_w$ with respect to $t_{1/2}^{LDL}$. For convenience, double $x$ and $y$ scales are given in **b**, **c**. The temperature dependence of $t_{1/2}^{LDL}$ is shown in an Arrhenius plot in the inset of **c**. The error bar in $t_{1/2}^{LDL}$ indicate the parameter uncertainty obtained from the fits using least-squares minimisation. The green line represents an Arrhenius fit with an activation energy of $410 \pm 20\,kJ/mol$ ($\approx 98\,kcal/mol$ or $\approx 4.2\,eV$). The horizontal dashed line in **b**, **c** indicates the boundary between stage-I and stage-II. Source data are provided as a Source Data file.

fused LDLs begin to form lower-order aggregates[50,51]. Considering the structural similarity of human-LDL and yolk-LDL, we anticipate that during stage-I, yolk-LDLs undergo a similar type of fusion and aggregation following the RLA mechanism. During stage-I, the correlation length exponentially increases from ≈50 nm, and the growth slows down when the $\zeta$ approaches ≈200 nm, thus showing an overall sigmoidal behaviour in a log-log plot.

To capture these changes, $\zeta$ curves are modelled using a sigmoidal function as explained in the Methods (Eq. (5)). Interestingly, the characteristic time $t_{1/2}^{LDL}$ extracted from the sigmoidal fits follows Arrhenius behaviour as displayed in the inset of Fig. 3c. The estimated activation energy, $E_a = 410 \pm 20\,kJ/mol$ ($\approx 4.2\,eV$), is in good agreement with that reported for human-LDL aggregation[51]. Further, a master curve (Fig. 3c) is obtained upon normalising the data presented in Fig. 3b using $t_{1/2}^{LDL}$, implying an identical aggregation mechanism with a temperature-dependent reaction rate. However, there is a small deviation of $\zeta$ values for $T = 100\,°C$ from the overall sigmoidal behaviour in stage-I and a slight deviation from the Arrhenius behaviour. We anticipate that there could be some additional effects due to the fast evaporation of water in egg yolk at the boiling point of water $100\,°C$.

The exponential growth stage is followed by a power-law growth (stage-II) $\zeta \sim t_w^\mu$ with an exponent $\mu = 0.27 \pm 0.04$. This exponent is in good agreement with values observed in colloidal aggregations[52], and liquid-liquid phase separation[53], where the particles/clusters diffuse and coalesce upon collisions. Note that stage-II is absent in the aggregation kinetics of human-LDLs in dilute solutions[51], hence the high concentration of yolk-LDLs could be the reason behind the presence of stage-II in egg yolk. In addition, we find the emergence of a second broad peak at $q = 1.8\,nm^{-1}$ in $I(q)$ collected in SAXS geometry (Supplementary Fig. 19) indicating the formation of intra-aggregate structures. We speculate that the self-assembly of LDL constituents results in such structural organisations inside the aggregates. The observed peak position is comparable to that reported in the literature for micellar structures[54], indicating the possibility of the formation of tiny micellar structures within the large aggregates utilising the phospholipids of ruptured LDLs, but future investigations are required to confirm this hypothesis.

A meaningful physical representation of the non-equilibrium process necessitates a simultaneous understanding of kinetics and dynamics. In the following sections, we focus on the microscopic dynamics of egg yolk and its fractions.

**a**

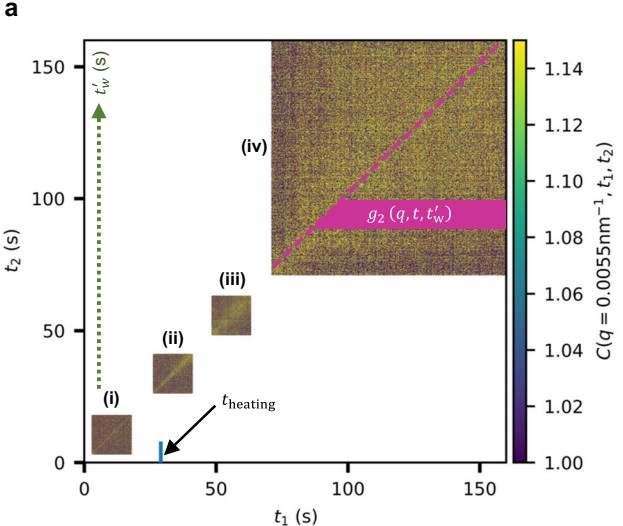
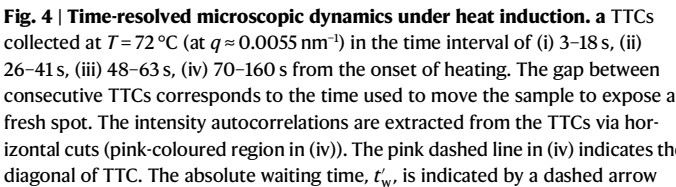

**b**

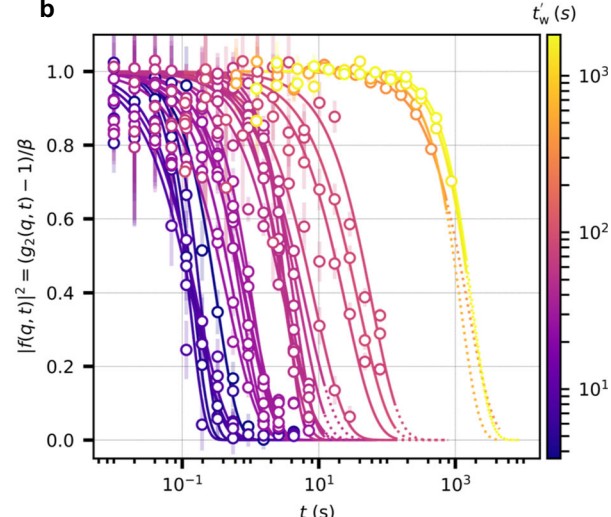

**Fig. 4 | Time-resolved microscopic dynamics under heat induction. a** TTCs collected at $T = 72\,°C$ (at $q \approx 0.0055\,nm^{-1}$) in the time interval of (i) 3–18 s, (ii) 26–41 s, (iii) 48–63 s, (iv) 70–160 s from the onset of heating. The gap between consecutive TTCs corresponds to the time used to move the sample to expose a fresh spot. The intensity autocorrelations are extracted from the TTCs via horizontal cuts (pink-coloured region in (iv)). The pink dashed line in (iv) indicates the diagonal of TTC. The absolute waiting time, $t'_w$, is indicated by a dashed arrow running along the $y$ axis. The short blue vertical line close to the $x$ axis represents the time ($t_{heating}$) taken to reach 72 °C from the initial temperature of 22 °C. **b** $|f(q,t)|^2$ extracted from the TTCs via horizontal cuts at different $t'_w$ as indicated by the colour bar. The error bars represent the standard error over TTC lines within a horizontal cut. The solid curves are fits using Eq. (3). The dotted curves represent the extrapolation of the fits for clarity. Source data are provided as a Source Data file.

## Monitoring the time-resolved dynamics

We extract the time-resolved sample dynamics via two-time correlation functions (TTCs)[22–26,39,53,55–59],

$$C(q, t_1, t_2) = \frac{\langle I_p(q, t_1) I_p(q, t_2) \rangle}{\langle I_p(q, t_1) \rangle \langle I_p(q, t_2) \rangle}, \quad (2)$$

where $I_p$ is the intensity at pixel $p$, $\langle ... \rangle$ denotes the average over pixels in a $q$-range of $q \pm \delta q$ ($q = 5.5 \pm 0.25\,\mu m^{-1}$ is used in this study), and $t_1$ and $t_2$ are different experimental times. A typical TTC is shown in Fig. 4a. The absolute waiting time, $t'_w$, increases along $t_2$ and $t'_w = 0$ indicates the onset of heating. The relative time $t = |t_1 - t_2|$ increases away from the diagonal in the horizontal direction. The time-resolved intensity autocorrelation function, $g_2(q, t, t'_w)$ at different $t'_w$ are obtained from the TTCs via horizontal cuts ($C(q, t + t'_w, t'_w)$)[53,55,60,61] along $t_1$ as shown in Fig. 4a(iv). The $g_2(q, t, t'_w)$ are modelled using the relation[22],

$$g_2(q, t, t'_w) = 1 + \beta |f(q, t)|^2 = 1 + \beta | \exp[-(t/\tau)^\gamma] |^2, \quad (3)$$

where $f(q, t)$, $\tau$, $\beta$, and $\gamma$ are the intermediate scattering function, relaxation time, speckle contrast, and Kohlrausch-Williams-Watts exponent[62], respectively. Here, $\gamma$ determines the shape of the correlation function and provides information about the nature of the dynamics: a simple exponential decay ($\gamma = 1$) indicates simple diffusive dynamics, compressed ($\gamma > 1$) and stretched ($\gamma < 1$) exponential decays characterise non-equilibrium and heterogeneous dynamics[56].

Before examining the non-equilibrium processes in the thermally driven yolk, it is important to understand the equilibrium dynamics of egg yolk. The microscopic dynamics of yolk measured at 22 °C exhibits hyper-diffusive dynamics ($\tau$~$1/q$) with a compressed exponential decay ($1 < \gamma < 2$) as shown in Supplementary Fig. 20. Such anomalous diffusive dynamics have been predicted[63] and reported in crowded biological environments[64–67]. Therefore, we anticipate that the macro-molecular crowding in egg yolk is responsible for the observed non-diffusive dynamics at room temperature. In addition, the dynamics of yolk measured at different sample positions and from different yolk samples displays a range of relaxation times (Supplementary Fig. 20) caused by the heterogeneous nature of yolk.

## Time and temperature-dependent microscopic dynamics

Recent studies have shown that protein samples are sensitive to radiation damage[61], both in terms of structural changes and the possibility of beam-induced acceleration of the dynamics[68–70]. Therefore, we carefully investigated the effect of dose and dose rate on both structure and dynamics (see Supplementary Fig. 7-8) and identified fluence and dose regimes without detectable changes to structure and dynamics.

To capture the evolution of microscopic dynamics in egg yolk during in-situ heating, we collected several successive XPCS scans starting from the onset of heating. Figure 4a displays the temporal evolution of the TTCs at $T = 72\,°C$. The TTCs for other temperatures are provided in Supplementary Fig. 21-24. When the sample is heated from $T = 22\,°C$ (at $t'_w = 0$) to 72 °C, a very fast dynamics is observed in the early stages as illustrated by the thin yellow diagonal region in the TTC (Fig. 4a(i)). Following this, a slow-down in the dynamics is observed after $t'_w \approx 30$ s as depicted in Fig. 4a(ii). The final temperature is reached in $\approx 30$ s, and the system starts to equilibrate at 72 °C, as indicated by the dynamical transition from Fig. 4a(i) to Fig. 4a(ii). With further waiting, the sample shows a slow ageing behaviour in the time window of $\approx 30$–80 s. After $t'_w \approx 80$ s, we observe a pronounced transition from fast to slow dynamics, as indicated by the divergence of the yellow region along the diagonal of the TTC depicted in Fig. 4a(iv). In addition to the TTCs shown in Fig. 4a, the extraction of relaxation times as a function of $t'_w$ provides a more quantitative picture. This is accomplished by extracting $|f(q, t)|^2$ from the TTCs at various $t'_w$ as shown in Fig. 4b ($|f(q, t)|^2$ for other temperatures are provided in Supplementary Fig. 25, 26). Notably, a dynamic slow-down over four orders of magnitude (few seconds to few hours) was observable in our measurements.

## Discussion

The temporal evolution of the structural features ($Q$ and $\zeta$) suggests that the series of non-equilibrium events in heated egg yolk follow

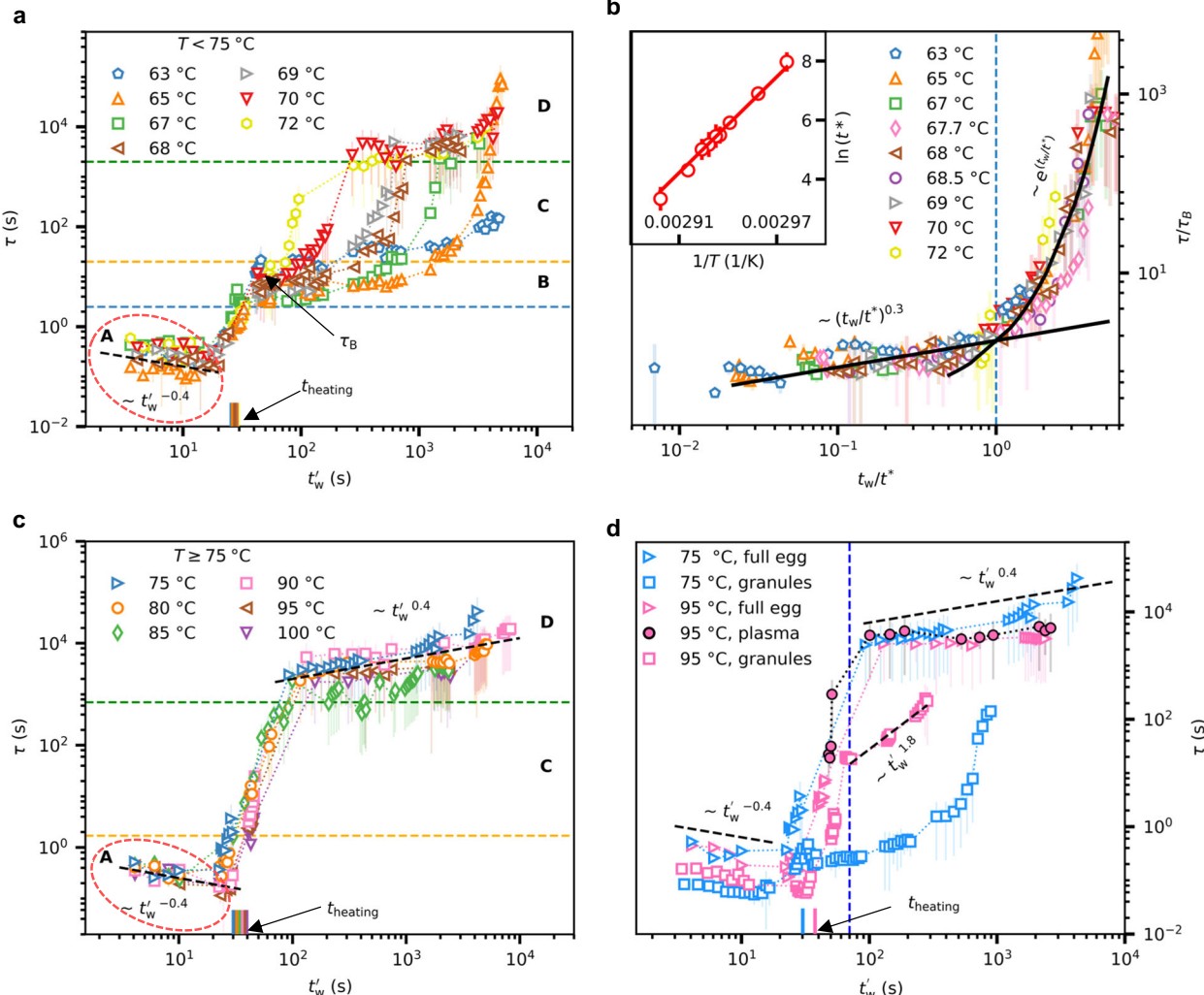

**Fig. 5 | Time and temperature-dependent microscopic dynamics. a** Temporal evolution of the relaxation time of egg yolk samples heated to temperatures in the range 63–72 °C. Four distinct dynamical regimes are identified. regime-A: accelerating motion of egg yolk-granules during heating, regime-B: equilibration of the system at the final temperature followed by protein denaturation and aggregation, regime-C: gelation of proteins, regime-D: ageing of the gel. The regime-A is denoted by the red dashed circle. The regimes B, C, and D are separated by horizontal dashed lines. The $t'_w$ and $\tau_B$ are the absolute waiting time and the equilibrium relaxation time from regime-B, respectively. The black dashed line indicates $\sim (t'_w)^{-0.4}$. **b** Normalised relaxation time with respect to $\tau_B$, as a function of $t_w$ (= $t'_w - t_{heating}$) normalised with respect to $t^*$ (sol-gel transition time). Here $t'_w$ and $t_{heating}$ are the absolute waiting time and the time taken to reach the final $T$, respectively. The two black curves before and after $t^*$ indicate a power-law ($\sim (t_w/t^*)^{0.3}$) and an exponential fit ($\sim \exp(t_w/t^*)$), respectively. An Arrhenius plot of $t^*$ with Arrhenius fit (solid line) is shown in the inset of **b**. The error bar in $t^*$ is estimated using the error in the fit parameters of power-law fits of normalised scattering invariant at low-$q$ and high-$q$ regimes as shown in Fig. 2 (see

Supplementary Note 6 for details). **c** Temporal evolution of the relaxation time of egg yolk heated to temperatures in the range of 75–100 °C. The regime-A is denoted by the red dashed circle. The regimes C and D are separated by horizontal dashed lines. The black dashed lines in regime-A and D indicate $\sim (t'_w)^{-0.4}$ and $\sim (t'_w)^{0.4}$ respectively. **d** Comparison of the relaxation time of egg yolk with egg yolk-plasma and yolk-granules (yolk-granule concentration is 910 mg/ml) at two temperatures as indicated in the legend. All $\tau$ are extracted for a $q$ value of 0.0055 nm$^{-1}$ except for the yolk-plasma sample, where the dynamical information was not accessible at this $q$ value. Hence we extrapolated $\tau$ at $q = 0.03$ nm$^{-1}$ to the value at $q = 0.0055$ nm$^{-1}$ using a $\tau \propto 1/q$ relationship. Details are provided in the Supplementary Note 9. The blue vertical line indicates $t'_w = 70$ s. The black dashed lines from bottom to top represent $\sim (t'_w)^{-0.4}$, $\sim (t'_w)^{1.8}$ and $\sim (t'_w)^{0.4}$. **a, c, d** The coloured short vertical lines close to the $x$ axis represent the time ($t_{heating}$) taken to reach a temperature $T$ from the initial temperature of 22 °C. The error bars in $\tau$ indicate the parameter uncertainty obtained from the fits using least-squares minimisation. Source data are provided as a Source Data file.

rather generic spatio-temporal scaling laws. To acquire a deeper understanding of these processes, we extract the dynamical information —the relaxation time, $\tau$—and examine its evolution as a function of $t'_w$ (= $t_w + t_{heating}$). Interestingly, the temporal evolution of $\tau$ (Fig. 5a,c) displays multiple dynamical regimes, which depend on temperature. At low and high temperatures the dynamics display four and three distinct regimes, respectively. We first focus on the dynamical behaviour at low temperatures (63-72 °C). As depicted in Fig. 5a, when heated from room temperature, a steady decrease in $\tau$ ($\sim (t'_w)^{(-0.4)}$) is observed until the set temperature is reached (regime-

A, red dashed circle in Fig. 5a). It has to be noted that the relaxation times are extracted for $q \approx 0.0055$ nm$^{-1}$, and the corresponding length scale ($\frac{2\pi}{q} \approx 1.1$ μm) is comparable to the size of the yolk-granules. Hence, we assume that the intensity fluctuations at this $q$ carry the information about the motion of the yolk-granules, and the initial fast dynamics are attributed to the movement of the granules caused by an increase in their kinetic energy. Another significant aspect of relaxation times in regime-A is the observed spread in $\tau$ between different temperatures, which we assign to the inherent heterogeneity in the sample.

Looking at Fig. 5a, after $t'_w \approx 30$ s, a quick transition from fast to slow dynamics is detected (regime-A to regime-B). On comparing this dynamical transition time with $t_{heating}$, it is clear that the observed slow-down implies the start of equilibration of the system at the final temperature. After $\approx 45$ s, the system is equilibrated and a gradual slowing-down is seen right after the equilibration in regime-B. It is interesting to note that this is the same time window in which we see the progressive increase in $Q/Q_0$ (Fig. 2b) until $t^*$. While the structural parameter $Q$ increases as $\sim t_w^{0.04}$, the dynamical parameter $\tau$ increases as $\sim t_w^{0.3}$ until $t^*$ (Fig. 5b). The strong correlation between structure and dynamics confirms that the probed $\tau$ in regime-B represents the collective dynamics in the egg yolk during protein aggregation. This is in good agreement with the notion that XPCS as a coherent scattering technique reflects the collective diffusion of the yolk-granules which is mirroring inter alia the viscosity of the denaturing protein environment. This hypothesis is verified later in this section when the dynamics of the full egg yolk and pure yolk-granules are compared. During regime-B, the egg yolk proteins unfold their native structure, exposing the buried hydrophobic/sulfhydryl groups and form aggregates via covalent and non-covalent bonds[44].

After regime-B, we observe a dynamical transition from power-law behaviour to exponential slow-down (regime-C), in which the time of transition decreases with increasing $T$. As expected, protein denaturation leads to gelation, which is manifested by a rapid exponential dynamical slow-down[68,71–74] indicated by an increase in $\tau$ by several orders of magnitude (regime-C). The sol-gel transition is characterised by a viscosity increase[75], and the viscosity is directly linked to $\tau$. Thus the significant slow-down in regime-C confirms the development of yolk gel. This observation is also supported by the exponential increase in the apparent viscosity of egg yolk measured using viscometry (see Supplementary Fig. 33). Remarkably, we find that at low temperatures ($T < 75$ °C) the onset of the exponential slow-down of the dynamics is identical to the $t^*$ obtained from the evolution of $Q$ from the structural analysis. The fast structural changes characterised by $Q/Q_0 \sim t_w^{0.2}$ after $t^*$ (Fig. 2c) reflect the build-up of the three-dimensional cross-linked network of protein aggregates. The TTS relationship of these non-equilibrium processes is evidenced by the collapse of $\tau$ onto a master curve upon normalising $t_w$ with $t^*$ as shown in Fig. 5b. Consequently, $t^*$ represents the transition point of protein denaturation and aggregation characterised by power-law dynamical slow-down (regime-B) to protein gelation characterised by rapid exponential slow-down (regime-C).

Remarkably, the $t^*$ values are in good agreement with sol-gel transition time extracted from in-situ viscometry measurements (see Supplementary Fig. 33). The $t^*$ values reported here are slightly lower than the sol-gel transition time estimated in the literature[8] using rheology, but the overall temperature trend is in good agreement. We speculate that the sol-gel transition time from rheology[8] would be even more close to $t^*$ if the Winter-Chambon criterion[76] is used for the estimation of sol-gel time as shown in[9]. The temperature dependence of $t^*$ follows an Arrhenius relationship[8,51],

$$\ln(t^*) = \frac{E_a}{RT} + A, \qquad (4)$$

where $R$ is the universal gas constant, $T$ is the temperature in Kelvin, and $A$ is a constant. The Arrhenius plot of $t^*$ is shown in the inset of Fig. 5b and the activation energy extracted from the fit is $450 \pm 20$ kJ/mol ($\approx 108$ kcal/mol or 4.7 eV). This high activation energy was reported earlier for the denaturation of bovine serum albumin (BSA)[8,77], human-LDLs[51] and other proteins[41,75,78,79]. The hen egg $\gamma$-livetin and yolk-LDL are structurally similar to BSA and human-LDL, respectively[8]. From this comparison, we deduce that the yolk livetins and apolipoproteins from LDLs are the major contributors to the heat-induced gelation of egg yolk at $T < 75$ °C. In the final stage (regime-D) the dynamics is evolving

very slowly, exhibiting an overall shallow power-law behaviour, which is suggestive of the ageing of a gel[80,81]. During this stage, the gel network stabilises by local reorganisation and the release of trapped stresses.

Next, we focus on the evolution of the dynamics at high temperatures 75 °C ≤ $T$ ≤ 100 °C, as depicted in Fig. 5c. Contrary to lower temperatures, the relaxation dynamics at high temperatures exhibits three dynamical regimes (A, C, D). At high temperatures, the protein gelation is initiated as soon as the final $T$ is reached. Consequently, there is no time for the system to equilibrate, hence regime-B is bypassed. This also implies that protein unfolding occurs within a fraction of a second at high temperatures. Another key consequence of this is the collapse of all curves onto a master curve in regime-C indicating approximately the same dynamical evolution. Following the gel formation, the ageing of the gel is observed in regime-D with an overall power-law behaviour ($\sim t_w^{0.4}$) which is identical to that found at low temperatures. Interestingly, the intermittent regions where $\tau$ deviates from the power-law dependence are indicative of stress relaxation[82–84].

In order to explicitly determine the contribution of the two egg yolk fractions: plasma and granules to the microscopic dynamics, we performed XPCS measurements on these fractions obtained from egg yolk (see Methods for preparation of yolk fractions). The extracted $\tau$ values of these fractions are compared with egg yolk at identical $T$, as shown in Fig. 5d. We first compare the dynamics of pure yolk-granules and egg yolk at $T = 75$ °C. The initial $\tau$ of granules at $t'_w \approx 4$ s is lower than that of full yolk, which is possibly caused by the higher water content of the granule solution (concentration of 910 mg/ml) and therefore lower viscosity.

After reaching the final $T$, the granules undergo equilibration similar to regime-B in Fig. 5a. Following this, there is a time interval until $t'_w \approx 500$ s, during which a power-law dynamical slow-down is observed. Interestingly, during this time interval, the full egg yolk sample shows signatures of protein gelation (exponential dynamical slow-down). This demonstrates the relative stability of the granules compared to other egg yolk-plasma proteins (livetins and apolipoproteins) and confirms our hypothesis that the granules mirror the evolution of the viscosity of the surrounding environment for $T < 75$ °C. It has to be noted that, though the yolk-granules seem to be relatively stable, the power-law behaviour of $\tau$ until $t'_w \approx 500$ s followed by an exponential slow-down indeed implies the gelation of yolk-granules at longer $t'_w$ compared to that of yolk-plasma proteins. Further, on comparing the dynamics of yolk-granules at 75 °C and 95 °C, it is clear that the onset of gelation of yolk-granules occurs quickly at high temperatures, hence the measured dynamics represent the collective dynamics of egg yolk. The contribution of egg yolk proteins to the observed egg yolk gelation is further confirmed by comparing $\tau$ of full yolk, yolk-plasma, and yolk-granules at $T = 95$ °C, as depicted in Fig. 5d. While the egg yolk-plasma follows the same dynamical behaviour as that of full egg yolk during the whole experimental time window, yolk-granules deviate from this behaviour after $t'_w \approx 70$ s. Following this short time interval, a steep power-law slow-down ($\sim (t'_w)^{1.8}$) is observed for yolk-granules. This implies a rather complex, multi-stage gelation process of yolk-granules, which requires further investigation.

By combining the simultaneous structural and dynamic information, we generate a time-temperature phase diagram of the non-equilibrium events in a thermally driven egg yolk, as depicted in Fig. 6. The phase diagram consists of three main regions: protein denaturation-aggregation, gelation, and ageing, which are indicated using different colours. While the low-temperature part of the phase diagram is covered by these three non-equilibrium processes, the high-temperature regime is more complex with multiple non-equilibrium events (protein gelation and LDL aggregation) occurring simultaneously. The Arrhenius relationship along the sol-gel transition time (red dashed line) represents the boundary between protein denaturation-aggregation and gelation. Interestingly, by extrapolating

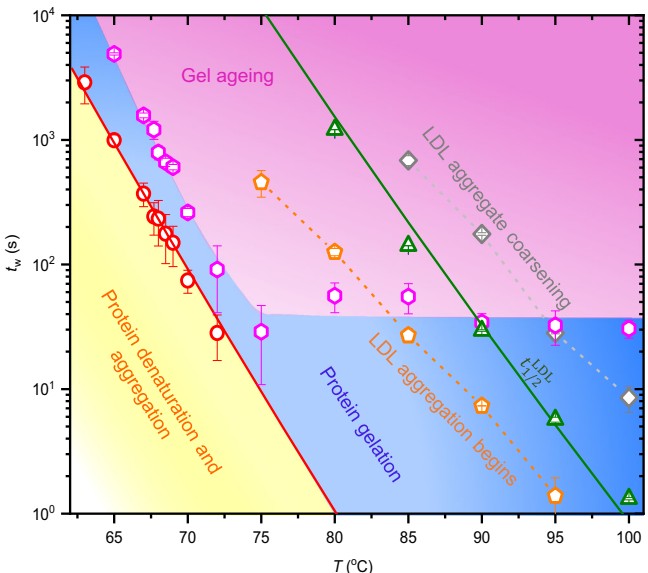

**Fig. 6 | Two-dimensional time-temperature phase diagram.** A master phase diagram showing different non-equilibrium processes that occur when egg yolk is cooked to temperatures in the range ≈63–100 °C. The phase diagram consists of three main regions, yellow: protein denaturation and aggregation, blue: protein gelation, and pink: gel ageing. red circle: $t^*$. The red line represents Arrhenius fit on $t^*$, which is extrapolated to higher temperatures. pink hexagon: Onset of egg yolk gel ageing ($t_g$), orange pentagon: onset of LDL aggregation, green triangle: $t_{1/2}^{LDL}$, grey diamond: onset of LDL aggregate coarsening (stage-II). The dotted lines are the lines connecting the data points. The green line represents Arrhenius fit on $t_{1/2}^{LDL}$. The information on LDL aggregation is derived from the structural analysis. The error bars in $t_g$, onset of LDL aggregation, and the onset of LDL aggregate coarsening indicate the error in determining the phase boundary from the available data points. Source data are provided as a Source Data file.

this curve to high temperatures, we can estimate the $t^*$ for $T \geq 75\,°C$, which was not resolvable in our experiments. The onset of gel ageing, $t_g$ is indicated by pink hexagon. Overall, $t_g$ follows a linear temperature dependence in the semi-log plot until $T = 75\,°C$. Interestingly, above 75 °C, $t_g$ is temperature-independent, indicating the minimum time (≈35 s) required for the collective interactions of lower-order protein aggregates to generate a three-dimensional gel network.

The microstructure of the cooked egg white, which is a soft gel, differs greatly from the grainy microstructure of the cooked egg yolk. Earlier studies from our group[25] on heat-induced gelation of egg white proteins describe protein denaturation and aggregation using a reaction-limited aggregation model coupled with an initial exponential increase in $Q$ during gelation. In contrast, a weak power-law increase in $Q$ is observed here. Furthermore, an exponential slow-down is observed during the gel ageing phase of egg white, in contrast to the power-law behaviour in egg yolk. The high concentration and diversity of constituents and pronounced multi-component interactions in egg yolk could be the reason behind these differences, especially for its grainy microstructure.

Incorporation of LDL aggregation stages extracted from $I(q)$ profiles, into the phase diagram reveal the coupling between protein denaturation and LDL aggregation (Fig. 6). The parallel Arrhenius fits (red and green lines) on $t^*$ and $t_{1/2}^{LDL}$ respectively point to an underlying connection between these two processes. We anticipate that the denaturation of apolipoproteins disrupts the structure of yolk-LDLs and promotes its aggregation. The approximately same $E_a$ for protein denaturation and LDL aggregation support this hypothesis. Additionally, the emergence of an LDL aggregation peak at a very long waiting time of ≈6000 s in the $I(q)$ for $T = 72\,°C$ (Supplementary Fig. 10)

confirms the consecutive nature of these events: protein denaturation-aggregation-gelation and LDL aggregation as a function of $t'_w$.

The time-temperature superposition principle is a concept widely used in polymers and glass formers to determine temperature-dependent mechanical properties of linear viscoelastic materials from known properties at a reference temperature[85]. Similarly, the master curves and the Arrhenius relationships obtained here can be utilised to predict the structure and dynamics of the system at different time and temperature combinations which are difficult to access in an experimental window. This means that the process/mechanism which leads to the formation of the structure is identical and the speed of this process depends on temperature.

In summary, we investigated the functional contribution of egg yolk constituents: proteins, LDLs, and yolk-granules to the formation of the grainy-gel microstructure of cooked egg yolk, by concurrently following structural and dynamical changes exploiting XPCS, for wide time-temperature combinations (0.1 s < $t_w$ < $10^4$ s and $63\,°C \leq T \leq 100$ °C). We find that the viscosity of the yolk is solely determined by the degree of gelation of yolk-plasma proteins, whereas the grainy microstructure of the gel is controlled by the extent of yolk-LDL aggregation. The protein denaturation-aggregation-gelation and LDL aggregation follow Arrhenius time-temperature relationships with a temperature-dependent reaction rate. The coupling between protein denaturation and LDL aggregation is supported by approximately similar activation energy (430 ± 40 kJ/mol) for these processes. The TTS indicates an identical mechanism underlying protein aggregation-gelation and LDL aggregation. The breakdown of TTS above 75 °C indicates a complex association of protein aggregates that results in the 3-dimensional gel network, which cannot be accelerated with increasing temperature. Consolidating the evidence, we generate a time-temperature phase diagram that aids to understand the coupling between nanoscale processes that take place during the heating of egg yolk. Our findings are not only relevant in food science and biomaterials but also benefit biophysics in relation to understanding the denaturation and aggregation processes in dense protein-lipid mixtures, on length scales ranging from nano- to micrometres in a time range of milli-seconds to hours.

## Methods

### In-situ coherent X-ray experiments

The XPCS measurements were performed at the beamline P10 of PETRA III, DESY, Hamburg, Germany. The data presented in this study were collected from different beamtimes, therefore the X-ray energy employed was 8.54 keV (for egg yolk and egg yolk-plasma) or 8.75 keV (for yolk-granules). The two-dimensional scattering pattern series were recorded using an EIGER X4M detector (pixel size = 75 × 75 μm²) mounted 21.2 m downstream of the sample stage. A fairly large X-ray beam of size 100 × 100 μm² enabled us to perform low-dose XPCS measurements below radiation damage limits (see Supplementary Note 4). In addition, the capillaries were displaced laterally between two measurements across the beam by 200 μm for consecutive measurements to avoid radiation damage due to over-exposure. Furthermore, the best configuration of silicon absorbers and total exposure time were chosen to reduce beam damage. Specifically, all measurements presented in this manuscript were performed using dose rates of 0.004, 0.025, and 0.046 kGy/s, and the total absorbed dose per scan is limited to 1 kGy (estimation of dose and dose rates are described in Supplementary Note 4). For in-situ heating measurements, the capillaries were mounted on a sample holder equipped with a temperature-controlled heating stage (Linkam Scientific Instruments Ltd., UK). Temperatures in the range 63 °C to 100 °C were selected for the measurement. More information on XPCS measurements and data analysis is provided in Supplementary Note 4.

Apart from XPCS measurements in USAXS geometry, we also performed SAXS measurements at the P10 beamline, PETRA III,

Hamburg and beamline BL2 of the DELTA synchrotron radiation source. The details are provided in Supplementary Note 4.

## Sample preparation

The hen egg was purchased from a local supermarket. The egg yolk was separated from the egg white and was washed slowly in deionised water and rolled on a filter paper to remove the excess albumin. The vitelline membrane, which separates egg yolk from egg white was punctured using a pipette tip and the yolk contents were extracted and filled in a falcon tube. The fresh hen egg yolk sample filled in a quartz capillary (diameter ≈1.5 mm), sealed with parafilm, was used for measurements. Different capillaries were used for each heating measurement. In order to separate the egg yolk fractions (plasma and granules), the egg yolk was centrifuged at $5270 \times g$ at a constant temperature of $20\,^{\circ}C$ for ≈36 hours. The top supernatant solution containing LDLs and the bottom granules were separated for reference XPCS measurements. The sample was stored at $T = 5\,^{\circ}C$ during the course of the experiments. Further details can be found in Supplementary Note 2.

## Modelling LDL aggregation

The stage-I of LDL aggregation kinetics is modelled using

$$\log_{10}(\zeta) = A2 + \frac{(A1 - A2)}{1 + \exp\left[\left(\log_{10}(t_w) - \log_{10}\left(t_{1/2}^{LDL}\right)\right)/k\right]}, \quad (5)$$

where $A1$ (=1.7) and $A2$ (=2.3) are the lower and upper plateau values respectively. The $\log_{10}(t_{1/2}^{LDL})$ is the midpoint of the fit and $k$ is a constant related to the slope of the curve at $\log_{10}(t_{1/2}^{LDL})$ through the relation, slope $= (A2 - A1)/(4k)$.

## Reporting summary

Further information on research design is available in the Nature Portfolio Reporting Summary linked to this article.

## Data availability

The processed data (scattering intensity profiles, intermediate scattering functions, and scattering invariant) of egg yolk samples for all temperatures have been deposited in Zenodo (https://doi.org/10.5281/zenodo.8202895). Any other data used in this study are available from the authors upon request. Source data are provided with this paper.

## Code availability

Custom Python scripts developed for the study are available at Zenodo (https://doi.org/10.5281/zenodo.8202895).

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

## Acknowledgements

We acknowledge DESY (Hamburg, Germany), a member of the Helmholtz Association HGF, for the provision of experimental facilities. Parts of this research were carried out at the beamline P10. Beamtime was allocated for proposals II-20210008 and II-20211600. The authors also thank the support of DFG (NFDI 40/1), BMBF (05K19PS1, 05K20PSa, 05K22PS1, 05K20VTA), and NFDI for this work. N.B. acknowledges the Alexander von Humboldt Foundation. M.P. thanks the DELTA machine group for providing synchrotron radiation for sample characterisation. A.R. acknowledges the Studienstiftung des deutschen Volkes. M.S.A. acknowledges funding by DAAD. M.M. acknowledges RESOLV, funded by the Deutsche Forschungsgemeinschaft (DFG, German Research Foundation) under Germany's Excellence Strategy—EXC-2033—Projektnummer 390677874. We acknowledge Dr. D.C.F. Wieland for providing access to the Bolin Gemini rotational HR nano-rheometer for viscometry measurements. Parts of this research were carried out at DESY NanoLab and we would like to thank Dr. T.F. Keller and A. Jeromin for their assistance in using scanning electron microscopy.

## Author contributions

C.G., F.S., and F.Z. designed the research. N.D.A., A.G., and M.P. prepared the egg yolk samples. N.D.A. and A.G. planned the measurements at PETRA III. M.P., N.D.A., and S.T. planned the measurements at DELTA. N.D.A., A.G., S.T., M.K., M.S.A., S.R., M.D.S., M.D., D.G., A.H., M.M., Ö.Ö., H.F.P., A.R., A.T., and M.P. conducted the experiment. F.W. and M.S. operated the P10 beamline at PETRA III. M.P. operated the beamline BL2 at DELTA. N.D.A. performed the data processing and analysis. N.D.A., C.G., F.Z., and A.G. discussed the XPCS data analysis with input from F.S., S.T., M.K., M.S.A., S.R., M.D.S., M.D., A.H., A.R., N.B., M.P., F.W., and M.S. The manuscript was written by N.D.A., and C.G. with input from all authors.

## Funding

## Competing interests

The authors declare no competing interests.
