## [Peer Review File · Nature Communications]

Exploring non-equilibrium processes and spatio-temporal scaling laws in heated egg yolk using coherent X-raysREVIEWER COMMENTS

Reviewer #1 (Remarks to the Author):

This paper uses X-ray photon correlation spectroscopy to study changes in molecular structure, molecular assembly, and molecular dynamics when egg yolks are heated. These changes in molecular states are considered to be closely related to the physical properties of the food. Although this is a complex sample with other components, the attribution of the signals has been relatively carefully experimented and discussed. The experiments seem to have been carried out precisely, and the results themselves are considered satisfactory, but there are not enough experimental results to support the discussion of food physical properties such as viscosity and texture. These physical properties should be verified by general food science measurement methods. More specifically, the results of experiments with macroscopic measures of viscosity and texture on samples heated by a process similar to the XPCS should be presented and compared to the XPCS results. I also recommend that the importance and necessity of simultaneous measurements of structure and dynamics be discussed in the discussion of experimental results. Significant revisions and improvements are considered necessary for publication. The other individual comments are below,

- 1) Page 3 line 17: The authors should be more specific and logical in describing why you can conclude from these results that it is an identical reaction pathway.
- 2) Page 6 line 7: How the Porod scattering is attributed to the egg yolk origin?
- 3) Page 6 line 8: How was the radius R evaluated? The sample in this study is complex with other components. If it is possible to identify the molecule being observed from a single scattering profile, the method should be more detailed.
- 4) Page 9 line 31 – Page 10 line 2 & page 11 line 37: The authors say that the yolk-granule plays the role of a tracer, but XPCS is data from coherent scattering and observes density fluctuations. As microscopic dynamics, the difference between self-diffusion and collective diffusion should be distinguished and discussed.

Reviewer #2 (Remarks to the Author):

Following the same group's investigation of on heat-induced gelation of egg white proteins, the manuscript present an exploration of the dynamics of egg yolk when heated.

Using XPCS the authors perform a detailed study of the multiple dynamical processes that happen in this complex mixture of proteins and lipids.

The results are well supported by experimental evidence.

I only have minor nitpicks with the manuscript:

- The plots of the two-time correlation functions (both in the main text and the supplementary material) seem to have some kind of strange interpolation. Please make sure to avoid it or use nearest neighbour interpolation.
- The text is at times repetitive. This makes it unnecessarily long. I think it could benefit from a further round of editing.
- The colors for t_g and onset of LDL aggregation in the caption of Fig. 6 don't seem to match the figure.

Overall I think it's solid work that only requires minor tweaks.

Reviewer #3 (Remarks to the Author):

The experiment design and data analysis resulting in the generic phase diagram is impressive. The accomplishment (a phase diagram of a dynamic process while out of equilibrium) is noteworthy as an experiment, particularly because one cannot uncook an egg yolk. Additionally, the volume of data and experiments across different beamlines illustrates the care, attention, and reproducibility of the methods. It is clear the researchers took great care in ensuring the validity of their results for comparison with others (temperature offset calibration) and eliminated x-ray beam induced effects.

Generally, the findings related to small angle scattering are further supported as they align with existing literature citations from domain specific journals. The quantitative analysis of the dynamic speckle data did provide more insight into the system that exists in the cited literature (and addresses short-comings or inconsistencies in the presented literature).

Related to human health (a high impact topic), the researchers introduced that LDL and protein agglomeration are important to a few human diseases, which are active areas of medical research. Generally, I would find some more closing discussion about how this study impacts those areas of active research very interesting. Or is it that the method here can be applied to study human disease related to proteins and LDL? What is the feasibility of this?

Overall, the manuscript includes some results comparisons with human and bovine LDL (both materials are highly relevant to medical science). However, these egg yolk processes are considerably above human body temperatures. With respect to protein aggregation-gelation, this general topic is important for therapeutics and medical devices; however, the temperature of the study is far above 37C. Is it that the researchers believe the Arrhenius-like behavior (Fig 6) may be extrapolated to find t_w at clinically relevant temperatures? If so, what is the impact of this finding – protein denaturation and aggregation on the timescale of hours before protein gelation?

My general impression of this work is positive. That said, there are some aspects of the manuscript that can be addressed.

Manuscript, pg 8, first paragraph: “The correlation length extracted from the $I(q)$ of yolk-plasma coincides exactly with that of the full yolk (Fig. S15 of the SI), confirming that LDL aggregation is not affected by the presence of yolk- granules in the surrounding medium.”

- Fig S15 does not include data from egg yolk-plasma for smaller t_w . There is a non-linearity in this region, and it is appropriate to add a small disclaimer to the above sentence. Something like “where measured”. However, reading more carefully I see that the egg-yolk plasma studies and full egg-yolk studies utilized the same incident energy (8.54 keV), as per the SI section 4. It is therefore confusing why data seem to be missing (Fig S15). The egg-yolk plasma studies seemed to have been performed to make the results clearer so it is puzzling why S15 is not fully populated for the egg yolk-plasma (open markers).

- The method used to extract the correlation length ($2\pi / q$ -of-peak) is not clearly described for Figs S14 – S16 and related Fig 3b, c . There are many beautiful plots as a function of correlation length, but I was unable to easily find a description of how q -peak was extracted from the scattering data. Were there fits to the peak? Was q^{-4} subtracted to find a simple maximum? Was Eq 1 applied (and over what ranges)? Or something else? Xana’s online documentation does not seem to contain this (or its default behavior or the available options).

- Related to background subtraction, SI Fig S4 doesn’t show the orange line for the raw egg-yolk data. I assume it is underneath the green. I suggest increasing `matplotlib.pyplot.plot lw` for the orange data.

Generally, this opens a curiosity I had while reading the entire paper and SI. I assume pure egg-yolk plasma was studied later to clear up some confusion, but it doesn’t look like Fig S17b is a simple

component of S17a. Nor does 16a look like a subcomponent of S14c. Understandably, these biomolecular mixtures are very challenging, but some further illustration, discussion, and/or demonstration seems lacking from my perspective.

The overall impact of this general curiosity is perhaps best illustrated by Fig. 3b and Fig. 3c, which concerns fitting the modeling describing the LDL aggregation kinetics (Eq. 5). I imagine that the peak position at smaller t_w falls outside the detection limit (t_w and/or q) (as shown by Fig 3a and supported by SI Fig14a). As such, it is understandable that there is only a portion of 75C data in the sigmoidal fit of Fig 3b.

- However, it is unclear to me why/how the dotted line representing $A_2(=2.3)$ was chosen. Some justification for the application of the model (Eq 5) is helpful. It seems that the complexity of this dual stage aggregation prevents fitting of the sigmoidal "plateau". What is the uncertainty on this fit and how does it affect the phase boundaries? How was the global value of A_1 and A_2 selected.

- As alluded to earlier, I appreciate that the 100C data are shown, but a large chunk of small t_w are missing (Fig S15). This brings some uncertainty to 100C behavior. Some discussion about the impact of this on the Fig 3 findings seems appropriate, especially considering what seems like some VERY SMALL deviation from Arrhenius behavior at 100C (Fig 3c insert) and the chunk of missing points (Fig 3b). Fig S14f shows 100C to be very complex for the whole egg-yolk. Including the data of the pure egg-yolk plasma at 100C in figure S16 would be helpful in evaluating the results of Fig 3b,c. Is there a possibility of an additional phase or mechanism at 100C? (Some of the 100K in Fig 3c not obeying Type I or Type II).

- Finally, are all data shown in S15 included in Fig 3b and Fig 3c? And are the data in Fig S17 included in Fig3b, c (or is the resolution here not appropriate)?

Manuscript, pg 12, Fig 5 and references to color for regions B and C.

- B and C are just varied shades of D (pink). The manuscript indicates that B and C should be yellow and blue, respectively.

Manuscript, pg 16, Fig 6 (and discussion of Fig 6 in the text).

- It is this reviewer's wish (request) to carry forward the colour assignments and labeling from Fig 5 to the Fig 6 and associated discussion. On pg 14, 2nd paragraph 2nd sentence: "The phase diagram consists of three main regions: protein denaturation-aggregation, gelation, and ageing, which are indicated using different colours":

- o protein denaturation-aggregation (Fig 5, "B", yellow)

- o gelation (Fig 5, "C", blue)

- o ageing (Fig 5, "D", pink)

- This is particularly important because region A of Figure 5 is green, which could be confused with the green in Figure 6.

Response to reviewers' comments on the manuscript "Exploring non-equilibrium processes and spatio-temporal scaling laws in heated egg yolk using coherent X-rays"

We thank all reviewers for their thorough and critical reading of our manuscript, their positive feedback and the constructive suggestions which helped us to refine the scientific content and readability of our manuscript. In the following, we provide a point-by-point response to all the comments raised by the reviewers including additional measurements as requested by reviewer-1. Our responses are highlighted in blue and the changes in the manuscript are highlighted in magenta. Further, the changes in the manuscript and supporting information (SI) are highlighted in yellow in the revised manuscript and SI respectively.

REVIEWER COMMENTS

Reviewer 1 (Remarks to the Author):

Reviewer's comment 1: This paper uses X-ray photon correlation spectroscopy to study changes in molecular structure, molecular assembly, and molecular dynamics when egg yolks are heated. These changes in molecular states are considered to be closely related to the physical properties of the food. Although this is a complex sample with other components, the attribution of the signals has been relatively carefully experimented and discussed. The experiments seem to have been carried out precisely, and the results themselves are considered satisfactory, but there are not enough experimental results to support the discussion of food physical properties such as viscosity and texture. These physical properties should be verified by general food science measurement methods. More specifically, the results of experiments with macroscopic measures of viscosity and texture on samples heated by a process similar to the XPCS should be presented and compared to the XPCS results.

Authors' response: We thank the reviewer for the thorough evaluation and constructive comments which helped us to refine the manuscript.

We also thank the reviewer for bringing in the discussion of comparing our XPCS results with food physical properties such as macroscopic viscosity and texture. Indeed, methods such as rheology, viscometry, and/or microscopy measurements are often utilized to assess the physical properties of food [1, 2]. Therefore, we performed additional systematic in-situ viscometry and ex-situ scanning electron

microscopy (SEM) measurements and discuss the results in the following,

- **Comparison of apparent viscosity from viscometry and relaxation time from XPCS**

The in-situ viscometry measurements were performed using a Bolin Gemini rotational HR nano-rheometer. The egg yolk sample was heated to a set temperature T from $20\text{ }^{\circ}\text{C}$ and the apparent viscosity η at a shear rate of 10 s^{-1} was measured in parallel. This shear rate is within the reported ranges for swallowing shear rates [3] and hence can be directly correlated with the mouth feel of food. The in-situ measurements were performed in a temperature range of $64\text{--}69\text{ }^{\circ}\text{C}$. In Fig. R1 we compare the apparent viscosity at different temperatures. Interestingly, the temporal evolution of η displays four regimes similar to our XPCS data. Initially, the viscosity of egg yolk is almost constant with $\eta \approx 2\text{ Pa}\cdot\text{s}$ for $\approx 30\text{ s}$ (regime-A). During this time period, the temperature of the yolk increases from $20\text{ }^{\circ}\text{C}$ to set T . After reaching this set temperature, a linear increase in viscosity can be observed. This continues until η reaches a value between $20\text{ Pa}\cdot\text{s}$ (for $64\text{ }^{\circ}\text{C}$) to $40\text{ Pa}\cdot\text{s}$ (for $69\text{ }^{\circ}\text{C}$). After this, the change in the viscosity is still linear with time (regime-B) but with a slow rate until a characteristic time, t_{visco}^* , beyond which the viscosity starts to increase exponentially (regime-C). Clearly, this sharp increase in η indicates the sol-gel transition [4] and t_{visco}^* is the sol-gel transition time. Interestingly, we obtain also a master curve upon normalising t_w with t_{visco}^* (Fig. R1b), which is very similar to the master curve obtained for τ (Fig. R1c). Moreover, the sol-gel transition times, t^* from XPCS and viscometry (t_{visco}^*) are in very good agreement (Fig. R1d).

However, there are a few differences in the τ and η data in the initial waiting time after regime-A. In the XPCS data, we see a sudden transition of the system from regime-A to regime-B while there is a progressive change in the viscosity data. This may arise from the change in the thermal history of the sample. For the XPCS measurements, the samples are contained in a thin capillary of diameter $\approx 1.5\text{ mm}$, whereas for the viscometry measurements, the samples are spread on a plate and the time needed to reach the same temperature everywhere in the sample may be slightly higher than the XPCS case. This might be the reason behind the observed differences between regime-A and regime-B.

We can not resolve the viscosity changes during the sol-gel transition time at temperatures above $69\text{ }^{\circ}\text{C}$ as this transition is too fast and already occurring while the whole sample reaches the final temperature set point.

- **Comparison of texture/microstructure from SEM with XPCS results**

The properties of "food texture" are usually characterised by means of the mechanical and/or geometrical characteristic properties [5]. The grainy texture of food samples can be characterized by the size and shape of the sample's constituents [5] for example by using microscopy techniques [6]. Here, we employ ex-situ scanning electron microscopy to investigate the effect of temperature on the grainy microstructure of heated egg yolk.

Fig. R1: (a) Apparent viscosity η of egg yolk heated to temperatures in the range of 64-69 °C as a function of absolute waiting time. Regime-A, B, C and D are indicated with the background colours of green, yellow, blue and pink respectively (similar to Fig. 5a of the manuscript). (b) Normalised apparent viscosity with respect to $\eta_0 \approx \eta(t = t_{visco}^*)$, as a function of isothermal waiting time, t_w normalised with respect to t_{visco}^* . (c) Master plot of τ from XPCS. (d) Comparison of sol-gel transition time from XPCS and viscometry.

Fig. R2: SEM images of egg yolk collected at room temperature after being heated at (a) 90 °C, (b) 95 °C, and (c) 100 °C for 300 s. Inset shows the histogram of LDL aggregate diameter. (d) LDL aggregate correlation length ζ from XPCS and aggregate diameter from SEM as a function of temperature.

In Fig. R2(a-c) we show SEM images collected from egg yolk heated to temperatures in the range 90-100 °C for 300 s. The grainy microstructure of egg yolk is indeed apparent in the SEM images. The size of these aggregates increases with increasing temperature (Fig. R2d). Importantly, we observe that the overall evolution of sizes (from SEM) and correlation lengths (from XPCS) of the food microstructure as a function of temperature are in good agreement as depicted in Fig. R2d.

However, it was difficult to perform SEM imaging of yolk samples heated at lower temperatures for the same waiting time due to aggregate sizes being smaller than 200 nm. To resolve such small structures, we had to expose the sample for a long time (compared to imaging bigger aggregates at high temperatures) to the electron beam (focusing, optimization of contrast etc. is necessary to resolve smaller structures). This led to imaging artefacts at low temperatures preventing us from quantitative analysis. We did not use any chemical fixation techniques to improve the electronic contrast, as it can damage the lipid content of egg yolk [7].

The term "texture" was used in relation to the microstructure of heated egg yolk. Hence, we replace the term "texture" in the manuscript with "microstructure" to be more precise and to avoid confusion.

Changes in the manuscript: The comparison of XPCS results with viscometry and SEM are added to the SI sections "Comparison of apparent viscosity from viscometry and relaxation time from XPCS" and "Comparison of microstructure from SEM with XPCS results" respectively (page 30 to 33 of the SI). This results in additional references [26-30] in the SI. In addition, this comparison is briefly mentioned in the manuscript by adding the following sentences,

- page 13 line 351: "This observation is also supported by the exponential increase in the apparent viscosity of egg yolk measured using viscometry (see Fig. S33 of the SI)."
- page 13 line 363: "Remarkably, the t^* values are in good agreement with sol-gel transition time extracted from in-situ viscometry measurements (see Fig. S33 of the SI)."
- page 8 line 200: "Moreover, the grainy microstructure of egg yolk and dispersity is indeed apparent in the SEM images (Fig. S34 of the SI) of heated egg yolk."
- page 8 line 209: "Remarkably, the overall evolution of correlation lengths from XPCS and aggregate sizes from SEM as a function of temperature are in good agreement as depicted in Fig. S34d of the SI."

The word "texture" is replaced by "microstructure".

- page 1 line 22: The soft-grainy microstructure of cooked egg yolk ...
- page 1 line 25: ... desired microstructure.
- page 1 line 30: ...whereas the grainy-gel microstructure is controlled ...
- page 2 line 43: ... the final gel microstructure is the result of a series ...

- page 3 line 98: ... nanoscale structure formation ...
- page 15 line 441: The microstructure of the cooked egg white, which ...
- page 15 line 442: ... the grainy microstructure of the cooked ...
- page 16 line 450: ... its grainy microstructure.
- page 16 line 471: ... the formation of the grainy-gel microstructure ...
- page 16 line 476: ... microstructure of the gel is controlled ...

Reviewer's comment 2: I also recommend that the importance and necessity of simultaneous measurements of structure and dynamics be discussed in the discussion of experimental results.

Authors' response: We thank the reviewer for this suggestion. In the revised manuscript, the following discussion is added to indicate the importance of simultaneous measurements of structure and dynamics,

The heat-induced gelation of egg yolk is the result of a series of out-of-equilibrium processes coupled by a hierarchy of length and time scales. While the microstructure of the system changes from a protein-lipid solution to a soft grainy gel network via protein gelation and LDL-aggregation, the viscosity is expected to show an exponential increase at this transition [8]. In the context of food science, this indicates the onset of changes in texture and correlates with the mouth feel of food. In general, these processes are also relevant for the fundamental understanding of nano- to micro-scale structure formation in concentrated protein/lipid systems. This implies that a complete understanding of these complex non-equilibrium processes necessitates a simultaneous understanding of changes in the structure and dynamics of its components [9, 10]. XPCS is a potential solution to accomplish this goal.

Changes in the manuscript: The above paragraph is added to the manuscript at the beginning of the "Results" section (page 3 line 100). This results in two additional references in the manuscript,

[34] Sankaran, J. et al. Simultaneous spatiotemporal super-resolution and multiparametric fluorescence microscopy. *Nature Communications* 12 (1), 1748 (2021).

[35] Lindorff-Larsen, K., Best, R. B., DePristo, M. A., Dobson, C. M. Vendruscolo, M. Simultaneous determination of protein structure and dynamics. *Nature* 433 (7022), 128–132 (2005).

Reviewer's comment 3: Significant revisions and improvements are considered necessary for publication. The other individual comments are below,

1) Page 3 line 17: The authors should be more specific and logical in describing why you can conclude from these results that it is an identical reaction pathway.

Authors' response: We thank the reviewer for bringing the discussion on the "identical reaction pathway" in egg yolk. The time-temperature superposition (TTS) of the structural parameters Q and ζ and the normalized dynamical parameter τ/τ_B including the Arrhenius behaviour of the corresponding characteristic times scales ($t_{LDL}^{1/2}$ and t^*) indicate that (a) the mechanism behind the structural and dynamical changes in the egg yolk during heating is independent of the temperature and (b) the rate at which these changes occur in the system depends on temperature. To put it in another way, although the temperature is the rate-limiting factor, the fundamental mechanisms behind denaturation-aggregation and gelation of yolk-plasma proteins and aggregation of LDLs are the same when the yolk is heated to temperatures in the range 63-100 °C.

The time-temperature superposition principle is a concept widely used in polymers and glass formers to determine temperature-dependent mechanical properties of linear viscoelastic materials from known properties at a reference temperature [11]. In this context, the instantaneous relaxation modulus as a function of time is shifted by a temperature-dependent scale factor and a master curve can be produced by applying a shift operation to various temperatures. Such property is widely used to predict the relaxation modulus of polymers at different temperatures using the temperature dependence of shift factors. Similarly, here we obtain a master curve for the structural and dynamical parameters of the system while egg yolk undergoes gelation. The Arrhenius relationship of characteristic time scales underlying the TTS can be utilised to predict the structure and dynamics of the system at different time and temperature combinations which are difficult to access in an experimental window. For example, to find the structural and dynamical changes in egg yolk at 60 °C until the final gel is completely formed, we need to perform an experiment for ≈ 4 hrs. Instead, the master curves and Arrhenius relationship of characteristic timescales can be used to find these parameters at such long time scales and temperatures below and close to 63 °C. This means that the process/mechanism which leads to the formation of the structure is identical and the speed of this process depends on temperature.

We understand that the word "identical reaction pathway" may be confusing for the reader as we do not discuss the chemistry of any biochemical reactions in the manuscript. Hence we replace the word "identical reaction pathway" with "identical mechanism" throughout the manuscript and SI to be more accurate.

Changes in the manuscript: The following sentences are added to the manuscript page 16 line 461:

"The time-temperature superposition principle is a concept widely used in polymers and glass formers to determine temperature-dependent mechanical properties of linear viscoelastic materials from known properties at a reference temperature [87]. Similarly, the master curves and the Arrhenius relationships obtained here can be utilised to predict the structure and dynamics of the system at different time and temperature combinations which are difficult to access in an experimental window. This means that the process/mechanism which leads to the formation of the structure is identical and the speed of

this process depends on temperature.”

This results in an additional reference in the revised manuscript:

[87] Urzhumtsev, Y. S. Time-temperature superposition. review. *Polymer Mechanics* 784 11 (1), 57–72 (1975)

The usage of the word ”identical reaction pathway” is changed to ”identical mechanism” in the revised manuscript.

- page 2 line 33: ...indicating an identical mechanism with a temperature ...
- page 3 line 88: ... implies identical mechanisms with temperature-dependent reaction rates.
- page 16 line 481: ... indicates an identical mechanism underlying protein ...

Reviewer's comment 4: 2) Page 6 line 7: How the Porod scattering is attributed to the egg yolk origin?

Authors' response: We thank the reviewer for the discussion on Porod scattering observed in egg yolk scattering intensity data. In Fig. R3(a) and (b) we show the schematics of the egg yolk components and the scattering data of yolk and its fractions: yolk-plasma and yolk-granules. Fig. R3b is same as Fig. S9 of the SI. The details for separating these fractions are provided in the SI. The Porod power law ($I(q) \propto q^{-4}$) is observed below $q \approx 0.04 \text{ nm}^{-1}$. The presence of the Porod behaviour at low- q in the full yolk and yolk-granules and the absence of the Porod law behaviour in the yolk-plasma data confirms that the Porod scattering in the full yolk can be attributed to the yolk-granules.

Fig. R3: (a) Schematic of different components of the hen egg yolk. Egg yolk can be separated into two fractions: yolk-plasma and yolk-granules via a centrifugation approach as described in the SI. The yolk-plasma is an assembly of a variety of proteins (livetins) and LDLs [12]. The LDLs, yolk-granules, and livetins constitute $\approx 66\%$, $\approx 22\%$, and $\approx 10\%$ of yolk dry matter, respectively [12–14]. LDLs are spherical core-shell molecules with an average diameter $\approx 30 \text{ nm}$ [14] and the egg yolk-granules are circular complexes with a diameter $\approx 0.3\text{--}2 \mu\text{m}$ [15–17]) made of LDLs, high-density lipoproteins (HDLs), and a protein called phosvitin [18, 19]. (b) Comparison of scattering profiles of full yolk, egg yolk-plasma, and egg yolk-granules. The USAXS profiles and SAXS profiles are stitched together for full yolk and yolk-plasma samples. The curves are shifted along the y-axis for clarity.

Changes in the manuscript: A sentence on page 5 line 150 in the revised manuscript is rephrased to indicate the comparison shown in Fig. R3b:

"...from the surface scattering of micron-sized yolk-granules. This is further confirmed by comparing the $I(q)$ of yolk, yolk-granules and yolk-plasma as depicted in Fig. S9 (see SI)."

Reviewer’s comment 5: 3) Page 6 line 8: How was the radius R evaluated? The sample in this study is complex with other components. If it is possible to identify the molecule being observed from a single scattering profile, the method should be more detailed.

Authors’ response: The average radius of the yolk-granules, $R \approx 1 \mu\text{m}$ is taken from the literature [15–17]. The size of yolk-granules is reported in the literature via scanning electron microscopy and transmission electron microscopy imaging. In Table R1, we provide a list of literature works where the size of the yolk-granule was estimated.

literature	diameter of yolk-granule	shape of yolk-granule	technique
Chang et al. [15]	0.3– 1.64 μm	circular	SEM
Chang et al. [15]	1.0– 1.3 μm	-	Coulter Counter distribution
Bellaris et al. [16]	2 μm	circular	SEM
Xu et al. [17]	0.2– 6.4 μm	circular	TEM

Table R1: Shape and size of yolk-granule reported in the literature.

We assume that the absence of form factor oscillations at low- q (below $q \approx 0.04 \text{ nm}^{-1}$) could be due to the dispersity in the size of the yolk-granules as shown in Table R1. However, due to the spherical shape of these micron-sized yolk-granule complexes, we can still observe the Porod scattering for $q > \pi/R \approx 0.003 \text{ nm}^{-1}$ [20].

As described in response to comment 4, we showed that the scattering profile of the egg yolk-granule separated from the yolk also shows a Porod power law behaviour and hence the origin of the Porod law seen in the full yolk is due to the yolk-granules. We hope that the question on Porod scattering is clarified in response to the reviewer’s comments 4 and 5.

Changes in the manuscript: The above-mentioned references ([15–17]) are cited in the following sentences in the revised manuscript,

- page 5 line 148: It has to be noted that the radius of yolk granules is $R \approx 1 \mu\text{m}$ [29-31] and Porod scattering...
- Fig. 1 caption: The egg yolk-granules are circular complexes (diameter $\approx 0.3\text{-}2 \mu\text{m}$ [29-31]) made of LDLs ...

Reviewer's comment 6: 4) Page 9 line 31 – Page 10 line 2 and page 11 line 37: The authors say that the yolk-granule plays the role of a tracer, but XPCS is data from coherent scattering and observes density fluctuations. As microscopic dynamics, the difference between self-diffusion and collective diffusion should be distinguished and discussed.

Authors' response: We thank the reviewer for pointing out the difference between self and collective diffusion. Indeed, XPCS measures collective diffusion which is only for diluted samples equivalent to self-diffusion.

Comparing our results with viscometry results (Fig. R1), we confirm that the dynamics of the large yolk-granules are indeed mirroring the evolution of the viscosity of the surrounding environment. We nevertheless agree with the reviewer that the wording tracer may lead to confusion and thus we decided to rephrase the relevant sentence.

Changes in the manuscript: The rephrased sentences are

- page 13 line 336: ...confirms that the probed τ in regime-B represents the collective dynamics in the egg yolk during protein aggregation. This is in good agreement with the notion that XPCS as a coherent scattering technique reflects the collective diffusion of the yolk-granules which is mirroring inter alia the viscosity of the denaturing protein environment.
- page 14 line 407: This demonstrates the relative stability of the granules compared to other egg yolk-plasma proteins (livetins and apolipoproteins) and confirms our hypothesis that the granules mirror the evolution of the viscosity of the surrounding environment for $T < 75^\circ\text{C}$.

Reviewer 2 (Remarks to the Author):

Reviewer's comment 1: Following the same group's investigation of on heat-induced gelation of egg white proteins, the manuscript presents an exploration of the dynamics of egg yolk when heated. Using XPCS the authors perform a detailed study of the multiple dynamical processes that happen in this complex mixture of proteins and lipids. The results are well supported by experimental evidence.

Authors' response: We thank the reviewer for the thorough and critical reading of our manuscript and the positive suggestions which helped us to refine the manuscript.

Reviewer's comment 2: I only have minor nitpicks with the manuscript: - The plots of the two-time correlation functions (both in the main text and the supplementary material) seem to have some kind of strange interpolation. Please make sure to avoid it or use nearest neighbour interpolation.

Authors' response: We used a bicubic interpolation method to acquire a high-resolution TTC from its low-resolution counterpart [21]. This was done to improve the image quality for the publication. In Fig. R4, we compare the TTCs without and with different interpolations. We find that "bicubic" interpolation seems to work better for the visualization of the results. It is clear that the TTC image quality is improved after applying a "bicubic" interpolation. In general a "nearest neighbour interpolation" works well when the image is scaled up and this is not the case here. Hence, for producing better quality TTCs for publication purposes we use the "bicubic" interpolation method and this does not affect any calculations or estimation of intensity auto-correlation functions.

Fig. R4: A typical TTC from egg yolk sample with (a) no interpolation applied, (b) 'nearest neighbour' interpolation applied and (c) 'bicubic' interpolation applied.

Changes in the manuscript: In the revised manuscript, the following sentences are added to the Fig. 4 caption and SI,

Manuscript (Fig. 4 caption): "A bicubic interpolation is applied to TTCs to improve image quality"

SI page 7 line 125: "A bicubic interpolation is applied to all TTCs presented in the manuscript to improve the image quality"

Reviewer's comment 3: - The text is at times repetitive. This makes it unnecessarily long. I think it could benefit from a further round of editing.

Authors' response: We thank the reviewer for this suggestion. We have now carefully edited the manuscript and removed unnecessary repetitions.

Changes in the manuscript:

- page 5 line 150: The following sentence
"...from the surface scattering of micron-sized yolk-granules. A comparison of scattering profiles of full yolk, yolk-plasma, and yolk-granules given in Fig. S9 (see SI) confirms the scattering contribution of granules at low q values."
is changed to
"...from the surface scattering of micron-sized yolk-granules. This is further confirmed by comparing the $I(q)$ of yolk, yolk-granules and yolk-plasma as depicted in Fig. S9 (see SI)."
- page 7: removed the sentence "The TTS is suggestive of a common reaction underlying these non-equilibrium processes with a temperature-dependent reaction rate."
- page 14: removed the sentence "This means that above a threshold temperature ($T \approx 75^\circ\text{C}$), thermal energy can no longer accelerate the gelation, and at least $\approx 35\text{ s}$ is required for the assembly of lower-order protein aggregates into a three-dimensional cross-linked network."
- page 14 line 392: Combined the sentences "Following the gel formation, the ageing of the gel is observed in regime-D. The overall power-law dependence ($\sim t_w^{0.4}$) in regime-D is identical to that found at low temperatures."
to
"Following the gel formation, the ageing of the gel is observed in regime-D with an overall power-law behaviour ($\sim t_w^{0.4}$) which is identical to that found at low temperatures."

Reviewer's comment 4: - The colors for t_g and onset of LDL aggregation in the caption of Fig. 6 don't seem to match the figure.

Authors' response: We thank the reviewer for pointing out this error. We have fixed this error in the revised version of the manuscript.

Reviewer's comment 5: Overall I think it's solid work that only requires minor tweaks.

Authors' response: We thank the reviewer for recognizing the quality of our work. We have implemented the reviewer's suggestions in the revised manuscript, and we hope that the changes are satisfactory and that the reviewer recommends our work for publication.

Reviewer 3 (Remarks to the Author):

Reviewer's comment 1: The experiment design and data analysis resulting in the generic phase diagram is impressive. The accomplishment (a phase diagram of a dynamic process while out of equilibrium) is noteworthy as an experiment, particularly because one cannot uncook an egg yolk. Additionally, the volume of data and experiments across different beamlines illustrates the care, attention, and reproducibility of the methods. It is clear the researchers took great care in ensuring the validity of their results for comparison with others (temperature offset calibration) and eliminated x-ray beam induced effects.

Generally, the findings related to small angle scattering are further supported as they align with existing literature citations from domain specific journals. The quantitative analysis of the dynamic speckle data did provide more insight into the system that exists in the cited literature (and addresses shortcomings or inconsistencies in the presented literature).

Authors' response: We thank the reviewer for the careful and constructive reading of our work, and the helpful ideas which enabled us to enhance the clarity of the scientific discussion in the manuscript.

Reviewer's comment 2: Related to human health (a high impact topic), the researchers introduced that LDL and protein agglomeration are important to a few human diseases, which are active areas of medical research. Generally, I would find some more closing discussion about how this study impacts those areas of active research very interesting. Or is it that the method here can be applied to study human disease related to proteins and LDL? What is the feasibility of this?

Authors' response: We thank the reviewer for bringing the discussion on the feasibility of this method in the context of human diseases. First, we try to address the impact and limitations of our results in relation to human diseases.

- Though the overall structure of human-LDL and yolk-LDL are comparable, there are differences in the shape, size, and the number of apolipoproteins attached to the LDL surface [12, 22]. However, the exponential aggregation stage (stage-I) of yolk-LDL observed is comparable with the exponential growth phase of human-LDLs [23, 24]. Whereas the second power-law growth regime is not observed in other human-LDL studies [23, 24]. We assume that a high concentration of yolk-LDLs is the reason behind the second step observed in our study.
- The aggregation of human-LDLs is initiated by the denaturation and dissociation of apolipoprotein [23, 24]. Similarly, we also observe a coupling between protein denaturation and yolk-LDL aggregation (Fig. R12). The parallel Arrhenius fits (red and blue dashed lines) on t^* and $t_{1/2}^{\text{LDL}}$ respectively point to an underlying connection between these two processes. We anticipate that the denaturation of apolipoproteins disrupts the structure of yolk-LDLs and promotes its aggregation. The approximately same activation energy, E_a for protein denaturation and LDL aggregation support this hypothesis.
- However, a direct comparison of our results with human-LDL aggregation-related diseases (atherosclerosis) is not straightforward since
 - (a) the risk of atherosclerosis is observed when the concentration of human-LDL is above ≈ 2 mg/ml [25] in the blood which is much lower than the concentration of LDL in egg yolk, and
 - (b) in atherosclerosis, oxidative modification of human-LDL by free radicals leads to aggregation, and the difference between temperature-induced aggregation and oxidative aggregation needs to be investigated. These are beyond the scope of this article, but the future extension of this work.
- It has been shown in the literature that protein aggregation can occur via different mechanisms such as diffusion-limited or reaction-limited aggregation mechanisms and it is specific to a particular protein system, the concentration of the proteins, etc [26]. Moreover, protein or LDL aggregation in concentrated solutions is significantly more complex compared to that of dilute solutions due to viscosity effects, multi-body interactions, excluded volume effects, short-range interactions, etc. [26]. Therefore, we believe that investigation of specific protein systems related to human diseases is necessary to draw concrete conclusions on human health.

Feasibility of XPCS to study human diseases related to proteins and LDL:

The low-dose XPCS is a powerful tool that can be utilized to study human diseases related to proteins and LDLs provided that one must take good care of radiation damage of these biomolecules induced by X-rays. In the past, we have shown that low-dose XPCS can be utilized to understand the structure and microscopic dynamics underlying liquid-liquid phase separation of concentrated protein solutions induced by temperature [27, 28] and pressure [29]. LLPS has been identified as an important step in the amyloid fibril formation of proteins which is linked to human diseases [30]. Recently, we have also performed similar XPCS studies on aggregation kinetics of human-LDL and the overall aggregation trend is in good agreement with the literature [23, 24]. The data is not shown as it is an ongoing investigation.

In summary, in-line with the editor's comments, we believe that our current egg-yolk manuscript does not allow to make claims about the feasibility of such measurements in the context of human health. Therefore, while we are happy to discuss it in this response letter, we decided to not discuss it in the manuscript. This is for future work to decide.

Changes in the manuscript: None

Reviewer's comment 3: Overall, the manuscript includes some results comparisons with human and bovine LDL (both materials are highly relevant to medical science). However, these egg yolk processes are considerably above human body temperatures. With respect to protein aggregation-gelation, this general topic is important for therapeutics and medical devices; however, the temperature of the study is far above 37°C. Is it that the researchers believe the Arrhenius-like behaviour (Fig 6) may be extrapolated to find t_w at clinically relevant temperatures? If so, what is the impact of this finding – protein denaturation and aggregation on the timescale of hours before protein gelation?

Authors' response: We thank the reviewer for bringing the discussion on the extrapolation of the Arrhenius behaviour (Fig. 6) to clinically relevant temperatures. In the following, we discuss the difficulty of extrapolation of Arrhenius's behaviour to clinical temperatures.

- Though the Arrhenius equation has been able to describe the temperature-dependent aggregation or gelation behaviour of several protein systems [31–33], non-Arrhenius behaviours have been observed in a number of cases [34]. Also in some cases, some proteins show multiple stages of unfolding and aggregation and each stage has an activation energy associated with it [35]. This makes it difficult to generate a universal model for protein aggregation kinetics.
- The chances of deviation of Arrhenius behaviour at low temperatures can not be neglected given the statistics of non-Arrhenius behaviours reported in the literature [34]. Hence it is required to collect as many data points as close to 37 °C to make the extrapolation error-free. However, to test the validity of the Arrhenius behaviour of yolk-proteins at a temperature of 50 °C, we need to perform a continuous experiment for 3 weeks during which the sample itself can degrade (shelf-life of the egg is 3-5 weeks under refrigerated conditions and the degradation is caused by bacterial action).

In conclusion, we believe that an extrapolation of our results to clinically low temperatures is not so straightforward due to the uncertainties mentioned.

Changes in the manuscript: None

Reviewer's comment 4: My general impression of this work is positive. That said, there are some aspects of the manuscript that can be addressed.

Manuscript, pg 8, first paragraph: "The correlation length extracted from the $I(q)$ of yolk-plasma coincides exactly with that of the full yolk (Fig. S15 of the SI), confirming that LDL aggregation is not affected by the presence of yolk- granules in the surrounding medium."

- Fig S15 does not include data from egg yolk-plasma for smaller t_w . There is a non-linearity in this region, and it is appropriate to add a small disclaimer to the above sentence. Something like "where measured". However, reading more carefully I see that the egg-yolk plasma studies and full egg-yolk studies utilized the same incident energy (8.54 keV), as per the SI section 4. It is therefore confusing why data seem to be missing (Fig S15). The egg-yolk plasma studies seemed to have been performed to make the results clearer so it is puzzling why S15 is not fully populated for the egg yolk-plasma (open markers).

Authors' response: We apologize for this error that occurred from our side. In Fig. R5 we show the full profile of yolk and yolk-plasma.

Fig. R5: The correlation length ζ estimated from scattering profiles as a function of t_w for egg yolk (solid) and egg yolk-plasma (open) samples. The temperatures are mentioned in the figure legends.

Changes in the manuscript: The Fig. S17 of the revised SI is replaced with Fig. R5 (page 20 of the revised SI).

Reviewer’s comment 5: - The method used to extract the correlation length ($2\pi / q$ -of-peak) is not clearly described for Figs S14 – S16 and related Fig 3b, c . There are many beautiful plots as a function of correlation length, but I was unable to easily find a description of how q -peak was extracted from the scattering data. Were there fits to the peak? Was q^{-4} subtracted to find a simple maximum? Was Eq 1 applied (and over what ranges)? Or something else? Xana’s online documentation does not seem to contain this (or its default behavior or the available options).

Authors’ response: We thank the reviewer for pointing out this issue. For egg yolk samples, it was difficult to find the peak position from the raw data. Hence, we have subtracted the background of $I(q) = Cq^{-\delta}$ to extract the peak position as depicted in Fig. R6. After the subtraction of $I(q) = Cq^{-\delta}$ from the raw data, the LDL-aggregate peak was clearly visible. An example illustrating these steps for an egg yolk data is shown in Fig. R6. After the subtraction, we used the function `peak_prominences` from "scipy.signal" Python library to find the peak position as indicated by a red dot in Fig. R6. At room temperature, δ was equal to 4, but with heating, there was a slight change in the slope (Fig. S16 of the revised SI), hence for subtracting the power-law background in the low- q , δ was set free.

Whereas for yolk-plasma samples, where q_{peak} position was clearly visible due to the absence of Porod scattering intensity at low- q regime, we simply extracted the maxima using `peak_prominences` from "scipy.signal" Python library as illustrated in Fig. R7.

Fig. R6: (a) A typical $I(q)$ data of heated egg yolk sample with power-law ($I(q) = Cq^{-\delta}$) fit at low- q regime. (b) The scattering intensity after the subtraction of $Cq^{-\delta}$. The red dot in the plots indicates the q_{peak} .

Changes in the manuscript: The above-mentioned details and Fig. R6 and Fig. R7 are added to the SI subsection “Extraction of correlation length from scattering intensity” (page 17-18 of the revised SI).

Manuscript, page 8 line 206: ... we also extract the temporal evolution of the correlation length, $\zeta = 2\pi/q_{peak}$ (see SI for details).

Fig. R7: A typical $I(q)$ data of heated egg yolk-plasma sample with the q_{peak} position indicated with a red dot. Note that X and Y scales are linear.

Reviewer's comment 6: - Related to background subtraction, SI Fig S4 doesn't show the orange line for the raw egg-yolk data. I assume it is underneath the green. I suggest increasing matplotlib.pyplot.plot lw for the orange data.

Authors' response: We thank the reviewer for this suggestion. The line width of raw egg yolk data (orange line) is increased for better visibility as shown in Fig. R8

Fig. R8: Comparison of raw and DI water subtracted $I(q)$ of the egg yolk. The background is approximately two orders of magnitude less than the yolk data.

Changes in the manuscript: The Fig. S5 of SI is replaced with Fig. R8 (page 6 of the revised SI).

Reviewer’s comment 7: Generally, this opens a curiosity I had while reading the entire paper and SI. I assume pure egg-yolk plasma was studied later to clear up some confusion, but it doesn’t look like Fig S17b is a simple component of S17a. Nor does 16a look like a subcomponent of S14c. Understandably, these biomolecular mixtures are very challenging, but some further illustration, discussion, and/or demonstration seems lacking from my perspective.

Authors’ response: We thank the reviewer for bringing up the discussion of egg yolk components. In Fig. R3b (Fig. S9 of the SI) we compare the scattering profile of full yolk with its two fractions yolk-plasma and yolk-granules. The USAXS and SAXS profiles are stitched together for full yolk and yolk-plasma samples. Unfortunately, we do not have the full scattering profile of yolk-granules. Nevertheless, from the available information, it is clear that the Porod scattering observed at the low- q regime (below $q \approx 0.04 \text{ nm}^{-1}$) comes from the yolk granules. From the structure factor peak position, we estimate an inter-particle distance of $\approx 30 \text{ nm}$, which is approximately equal to the reported size of yolk LDLs. Since $\approx 85\%$ of yolk-plasma is constituted by yolk-LDLs, we assume that this value is approximately equal to the diameter of LDLs.

- The data shown in Fig. S19 of the revised SI (Fig. S17 before revision) is collected from the DELTA beamline (details are given in the SI) and the q range accessible was $0.26\text{--}3.3 \text{ nm}^{-1}$ and hence the main structure factor peak position at 0.22 nm^{-1} could not be accessed here. We agree that there is a small change in the scattering intensity of the peak at 1.8 nm^{-1} between yolk and yolk-plasma. Nevertheless, the position of the peak is the same in both cases. This peak indicates the formation of intra-aggregate structures within the LDL aggregates.
- The Fig. S18a of the revised SI (Fig. S16a before revision) is collected in SAXS geometry at P10 beamline (PETRA III) and the accessible q range is $0.016\text{--}1.18 \text{ nm}^{-1}$, whereas Fig. S16c of the revised SI (Fig. S14c before revision) is collected in USAXS geometry at P10 beamline (PETRA III) and the accessible q range is $0.005\text{--}0.2 \text{ nm}^{-1}$. Nevertheless, we could see the aggregate peak in both images. To make this point clear, we compare the $I(q)$ profile from Fig. S18a of the revised SI (Fig. S16a before revision) and Fig. S16c of the revised SI (Fig. S14c before revision) at a waiting time of $t'_w \approx 190 \text{ s}$ as depicted in Fig. R9. It is clear that the aggregate peak position is approximately at the same q value in both cases, while the scattering profile of the yolk sample has a modulation of $I(q) \propto q^{-4}$ at low- q which originating from the Porod scattering of yolk-granules.

Fig. R9: Scattering profile of full egg yolk and egg yolk-plasma heated to $85 \text{ }^\circ\text{C}$ for $t'_w = 190 \text{ s}$.

Changes in the manuscript:

- A sentence on page 5 line 150 in the revised manuscript is rephrased to indicate the comparison shown in Fig. R3b:

"...from the surface scattering of micron-sized yolk-granules. This is further confirmed by comparing the $I(q)$ of yolk, yolk-granules and yolk-plasma as depicted in Fig. S9 (see SI)."

- The following lines are added to page 12 line 190 of SI:

"...From the structure factor peak position, we estimate an inter-particle distance of ≈ 30 nm, which is approximately equal to the reported size of yolk LDLs. Since $\approx 85\%$ of yolk-plasma is constituted by yolk-LDLs, we assume that this value is approximately equal to the diameter of LDLs."

- We realised that the q ranges of SAXS measurements are not explicitly mentioned in the SI. Hence, the following sentences are added to the SI page 4 line 96:

"The q -range of SAXS experiments at P10 beamline (PETRA III) is 0.016 - 1.18 nm^{-1} , and that for BL2 beamline (DELTA synchrotron radiation source) experiments is 0.26 - 3.3 nm^{-1} ."

Reviewer's comment 8: The overall impact of this general curiosity is perhaps best illustrated by Fig. 3b and Fig. 3c, which concerns fitting the modeling describing the LDL aggregation kinetics (Eq. 5). I imagine that the peak position at smaller t_w falls outside the detection limit (t_w and/or q) (as shown by Fig 3a and supported by SI Fig14a). As such, it is understandable that there is only a portion of 75C data in the sigmoidal fit of Fig 3b.

- However, it is unclear to me why/how the dotted line representing $A_2(=2.3)$ was chosen. Some justification for the application of the model (Eq 5) is helpful. It seems that the complexity of this dual stage aggregation prevents fitting of the sigmoidal "plateau". What is the uncertainty on this fit and how does it affect the phase boundaries? How was the global value of A_1 and A_2 selected.

Authors' response:

- The peak position at smaller t_w was difficult to extract since the corresponding q value falls outside the detection regime in USAXS q -range. This is the reason for the absence of 75 °C data until $t_w \approx 480$ s.
- A sigmoidal fit was used to follow Lu et al. [23], where aggregation of the human-LDL sample shows a sigmoidal-type aggregation behaviour.
- It is clear from the 75 °C data that there is an initial plateau region in the aggregation profile (marked using a blue circle in Fig. R10), which is not resolvable at high temperatures as the characteristic timescale increases exponentially with temperature (Arrhenius behaviour). Hence this plateau value of $\zeta = 50$ nm was chosen to fix $A_1=1.7(\log(50))$.
- Similarly, for 80 °C and 85 °C data, the exponential increase of ζ values slows down around $\zeta \approx 200$ nm (indicated by red circles in Fig. R10). Also, the transition from sigmoidal to power law behaviour seems to occur at this value. This is the reason for choosing $A_2=2.3 (\log(200))$.
- Therefore, the phase boundaries $\zeta = 50$ nm and $\zeta = 200$ nm are chosen following above two points.
- We agree that at high temperatures (> 90 °C) this plateau is almost invisible as the rate of the aggregation has increased exponentially with temperature (Arrhenius behaviour). Unfortunately, due to the limited time slot of the experiment, we could not collect data for more than ≈ 6000 s at 80 °C, which could have helped us to find the full curvature of this sigmoidal at very long waiting times.
- The light-coloured shaded region around the fitted curves in Fig. R10 shows the uncertainty of the fit with 95% confidence level.

Changes in the manuscript: The Fig. 3b of revised manuscript includes 95% confidence band. And the following sentence is added to the Fig. 3b caption of the revised manuscript

Fig. 3b caption: "The light-coloured shaded area around the solid sigmoidal fit lines represents the 95% confidence band."

Fig. R10: Fig. 3b of the manuscript re-plotted to show the 95% confidence band. The light-coloured shaded area around the solid sigmoidal fit lines represents the 95% confidence band. The red- and blue-coloured circles are used to highlight the A1 and A2 plateau of the sigmoidal feature respectively.

Reviewer's comment 9: - As alluded to earlier, I appreciate that the 100C data are shown, but a large chunk of small t_w are missing (Fig S15). This brings some uncertainty to 100C behavior. Some discussion about the impact of this on the Fig 3 findings seems appropriate, especially considering what seems like some VERY SMALL deviation from Arrhenius behavior at 100C (Fig 3c insert) and the chunk of missing points (Fig 3b). Fig S14f shows 100C to be very complex for the whole egg-yolk. Including the data of the pure egg-yolk plasma at 100C in figure S16 would be helpful in evaluating the results of Fig 3b,c. Is there a possibility of an additional phase or mechanism at 100C? (Some of the 100K in Fig 3c not obeying Type I or Type II).

Authors' response: We thank the reviewer for the careful evaluation of Fig. 3 and for bringing the discussion of 100 °C data.

- We deeply apologize for this error - missing some data points of 100 °C at low t_w - that occurred from our side. We have fixed this problem and show all data points of 100 °C in Fig. R5. Further, there are some unavoidable gaps in the data due to the following reasons,
 - In order to reduce the beam damage (due to overexposure to X-ray) in our XPCS measurement, we made sure that a sample spot is exposed for a total dose of < 1 kGy. When a sample location is shifted, some time (8 – 15 s) is often utilized for the sample motor movement.
 - Also sometimes when an absorber setting was changed between two scans, again 5 – 20s is lost for the motor movement of the absorber unit (we used two absorber settings for yolk samples with dose rates of 0.004 and 0.025 kGy/s).
 - This time gap is not at the same t'_w for all samples as we try to optimize the scan length depending on how fast or slow the changes are occurring in the system.

Further, we have refitted the sigmoidal curve on 100 °C data considering all data points as shown in Fig. R10 and produced it in the revised manuscript.

- Clearly, the stage-II of 100 °C data falls perfectly on the master curve in Fig. 3c. However, we agree with the reviewer that stage-I of 100 °C data seems to slightly deviate from the overall master curve and there is a small deviation of Arrhenius behaviour at 100 °C. A possible explanation is that part of the water that constitutes the egg yolk (50% weight fraction) boils. Nevertheless, we do not observe any such changes in the microscopic dynamics at 100 °C, which means that protein gelation seems to be unaffected. We again thank the reviewer for sharing this observation and we consider this issue for our future investigations. For the completeness of this manuscript, we have mentioned this observation in the revised manuscript.

Changes in the manuscript: Fig. 3b (same as Fig. R10) and Fig. 3c are revised by adding the full data of 100 °C. We have also refitted the sigmoidal curve on the full 100 °C data in Fig. 3b and the new $t_{LDL}^{1/2}$ (slightly less than the old one) is shown in Fig. 3c inset of the revised manuscript.

The following discussion is added to the revised manuscript to emphasize the deviation of 100 °C data in stage-I of LDL-aggregation.

page 9 line 234: "... with a temperature-dependent reaction rate. However, there is a small deviation of ζ values for $T = 100^\circ\text{C}$ from the overall sigmoidal behaviour in stage-I and a slight deviation from the Arrhenius behaviour. We anticipate that there could be some additional effects due to the fast evaporation of water in egg yolk at the boiling point of water 100°C ."

Reviewer's comment 10: - Finally, are all data shown in S15 included in Fig 3b and Fig 3c? And are the data in Fig S17 included in Fig3b, c (or is the resolution here not appropriate)?

Authors' response: All egg yolk data shown in Fig. S17 of the revised SI (Fig. S15 before revision) is included in Fig. 3b. As mentioned in the response to comment 7, the q range of the data shown in Fig. S19 of the revised SI (Fig. S17 before revision) is $0.26\text{--}3.3\text{ nm}^{-1}$ and the LDL-aggregate peak is not accessible here. Fig. S19 of the revised SI (Fig. S17 before revision) was included to show the intra-aggregate length scale of LDL aggregates, while Fig. 3b-c shows the inter-aggregate length scale.

Reviewer's comment 11: Manuscript, pg 12, Fig 5 and references to color for regions B and C.

- B and C are just varied shades of D (pink). The manuscript indicates that B and C should be yellow and blue, respectively.

Authors' response: We realise that this problem occurs if we try to open the manuscript pdf on Mac machines. Probably it is due to some compatibility issues between windows and Mac as we converted the four-panel figures from a pptx to pdf. But now we have fixed this problem and the yellow and blue regions are visible as shown in Fig. R11.

Fig. R11: Fig. 5 of the main manuscript

Reviewer's comment 15: Manuscript, pg 16, Fig 6 (and discussion of Fig 6 in the text)

- It is this reviewer's wish (request) to carry forward the colour assignments and labeling from Fig 5 to the Fig 6 and associated discussion. On pg 14, 2nd paragraph 2nd sentence: "The phase diagram consists of three main regions: protein denaturation-aggregation, gelation, and ageing, which are indicated using different colours": o protein denaturation-aggregation (Fig 5, "B", yellow) o gelation (Fig 5, "C", blue) o ageing (Fig 5, "D", pink)

- This is particularly important because region A of Figure 5 is green, which could be confused with the green in Figure 6.

Authors' response: We thank the reviewer for the careful note of the details and the constructive suggestion on the figure background colours to reduce the elements of confusion between Fig. 5 and Fig. 6. We have changed the colour codes of Fig. 6 as per reviewer's suggestion and reproduced in Fig. R12.

Fig. R12: Fig. 6 of the main manuscript. The colour of the protein denaturation and aggregation regime is changed from green to yellow. And colour of protein gelation is changed from sky-blue to blue to be consistent with colors shown in Fig. R11a

Changes in the manuscript: The Fig. 6 of the manuscript is replaced with Fig. R12 and the correct colour codes are mentioned in the Fig. 6 caption.

Authors' response: We hope that we were able to address all of the reviewer's comments satisfactorily so that the publication is recommended for the publication.

Additional changes:

- The following sentences are added to acknowledgement,

"We acknowledge Dr. D.C.F. Wieland for providing access to Bolin Gemini rotational HR nano-rheometer for viscometry measurements. Parts of this research were carried out at DESY NanoLab and we would like to thank Dr. T. F. Keller and A. Jeromin for their assistance in using scanning electron microscopy."

- The typo in the word "phosphitin" is corrected to "phosvitin" in the following locations: page 5 line 155, Fig. 1 caption, Fig. S1 caption.
- The following sentence is added to Methods section, page 17 line 525 "Different capillaries were used for each heating measurement."
- A section for "Data availability" and "Code availability" is added to the manuscript on page 18.
- The following line is added to the end of the caption of manuscript Fig. 2-6, "Source data are provided as a Source Data file."
- The details of error bars are indicated in figure captions as mentioned below.
- Fig. 2 caption: ...The error in Q is estimated by considering the standard error in $I(q)$. The error bars in Q/Q_0 represent the standard deviation estimated via error propagation.
- Fig. 2 caption: ...The error bar in t^* is estimated using the error in the fit parameters of power law fits at low- q and high- q regime.
- Fig. 3 caption: ... The error bars are obtained from the error in q_{peak} estimation (see SI).
- Fig. 3 caption: ...The error bar in $t_{1/2}^{\text{LDL}}$ indicate the parameter uncertainty obtained from the fits using least-squares minimization.
- Fig. 4 caption: ...The error bars represent the standard error over TTC lines within a horizontal cut.
- Fig. 5 caption: ...The error bars in τ indicate the parameter uncertainty obtained from the fits using least-squares minimization.
- Fig. 6 caption: ...The error bars in t_g , onset of LDL aggregation, and the onset of LDL aggregate coarsening indicate the error in determining the phase boundary from the available data points.

References

- [1] Gipsy Tabilo-Munizaga and Gustavo V Barbosa-Cánovas. Rheology for the food industry. *Journal of food engineering*, 67(1-2):147–156, 2005.
- [2] Michael H Tunick. Food texture analysis in the 21st century. *Journal of agricultural and food chemistry*, 59(5):1477–1480, 2011.
- [3] NR Pollen, Christopher R Daubert, P Prabhasankar, MA Drake, and ML Gumpertz. Quantifying fluid food texture. *Journal of texture studies*, 35(6):643–657, 2004.
- [4] Plinio Innocenzi et al. *The sol to gel transition*. Springer, 2016.
- [5] Alina Surmacka Szczesniak. Classification of textural characteristics a. *Journal of food science*, 28(4):385–389, 1963.
- [6] Jinping Dong and Var L. St. Jeor. Food microstructure techniques. In S. Suzanne Nielsen, editor, *Food Analysis*, pages 557–570. Springer International Publishing, Cham, 2017.
- [7] Kuo-Chiang Hsu, Wen-Hsin Chung, and Kung-Ming Lai. Histological structures of native and cooked yolks from duck egg observed by sem and cryo-sem. *Journal of agricultural and food chemistry*, 57(10):4218–4223, 2009.
- [8] César Vega and Ruben Mercadé-Prieto. Culinary biophysics: on the nature of the 6X° C egg. *Food Biophysics*, 6(1):152–159, 2011.
- [9] Jagadish Sankaran, Harikrushnan Balasubramanian, Wai Hoh Tang, Xue Wen Ng, Adrian Röllin, and Thorsten Wohland. Simultaneous spatiotemporal super-resolution and multi-parametric fluorescence microscopy. *Nature Communications*, 12(1):1748, 2021.
- [10] Kresten Lindorff-Larsen, Robert B Best, Mark A DePristo, Christopher M Dobson, and Michele Vendruscolo. Simultaneous determination of protein structure and dynamics. *Nature*, 433(7022):128–132, 2005.
- [11] Yu S Urzhumtsev. Time-temperature superposition. review. *Polymer Mechanics*, 11(1):57–72, 1975.
- [12] Marc Anton. Egg yolk: structures, functionalities and processes. *Journal of the Science of Food and Agriculture*, 93(12):2871–2880, 2013.
- [13] W J Stadelman and O J Cotterill. *Egg Science and Technology (4th ed.)*. CRC Press, Boca Raton, 1995.
- [14] Rainer Huopalahti, Marc Anton, Rosina López-Fandiño, and Rüdiger Schade. *Bioactive Egg Compounds*, volume 5. Springer, Berlin, 2007.
- [15] CM Chang, WD Powrie, and O Fennema. Microstructure of egg yolk. *Journal of Food Science*, 42(5):1193–1200, 1977.
- [16] Ruth Bellairs. The structure of the yolk of the hen's egg as studied by electron microscopy: I. the yolk of the unincubated egg. *The Journal of Cell Biology*, 11(1):207–225, 1961.

- [17] Lilan Xu, Yan Zhao, Mingsheng Xu, Yao Yao, Na Wu, Huaying Du, and Yonggang Tu. Changes in physico-chemical properties, microstructure, protein structures and intermolecular force of egg yolk, plasma and granule gels during salting. *Food chemistry*, 275:600–609, 2019.
- [18] Marc Anton, M Le Denmat, and Gilles Gandemer. Thermostability of hen egg yolk granules: Contribution of native structure of granules. *Journal of Food Science*, 65(4):581–584, 2000.
- [19] T Strixner, J Sterr, U Kulozik, and R Gebhardt. Structural study on hen-egg yolk high density lipoprotein (HDL) granules. *Food Biophysics*, 9(4):314–321, 2014.
- [20] Eugen Mircea Anitas. Small-angle scattering from mass and surface fractals. In Ricardo López-Ruiz, editor, *Complexity in Biological and Physical Systems*, chapter 10. IntechOpen, Rijeka, 2017.
- [21] Ankit Prajapati, Sapan Naik, and Sheetal Mehta. Evaluation of different image interpolation algorithms. *International Journal of Computer Applications*, 58(12):6–12, 2012.
- [22] Elena V Orlova et al. Three-dimensional structure of low density lipoproteins by electron cryomicroscopy. *Proceedings of the National Academy of Sciences*, 96(15):8420–8425, 1999.
- [23] Mengxiao Lu, Donald L Gantz, Haya Herscovitz, and Olga Gursky. Kinetic analysis of thermal stability of human low density lipoproteins: a model for LDL fusion in atherogenesis. *Journal of Lipid Research*, 53(10):2175–2185, 2012.
- [24] Mengxiao Lu and Olga Gursky. Aggregation and fusion of low-density lipoproteins in vivo and in vitro. *Biomolecular Concepts*, 4(5):501–518, 2013.
- [25] Wenrui Hao and Avner Friedman. The LDL–HDL profile determines the risk of atherosclerosis: a mathematical model. *PloS one*, 9(3):e90497, 2014.
- [26] Lucrèce Nicoud et al. Kinetics of monoclonal antibody aggregation from dilute toward concentrated conditions. *The Journal of Physical Chemistry B*, 120(13):3267–3280, 2016.
- [27] Anita Girelli et al. Microscopic dynamics of liquid-liquid phase separation and domain coarsening in a protein solution revealed by X-ray photon correlation spectroscopy. *Physical Review Letters*, 126(13):138004, 2021.
- [28] Anastasia Ragulskaya et al. Interplay between kinetics and dynamics of liquid–liquid phase separation in a protein solution revealed by coherent X-ray spectroscopy. *The Journal of Physical Chemistry Letters*, 12(30):7085–7090, 2021.
- [29] M. Moron et al. Gelation dynamics upon pressure-induced liquid–liquid phase separation in a water–lysozyme solution. *The Journal of Physical Chemistry B*, 126(22):4160–4167, 2022.
- [30] Susmitha Ambadipudi, Jacek Biernat, Dietmar Riedel, Eckhard Mandelkow, and Markus Zweckstetter. Liquid–liquid phase separation of the microtubule-binding repeats of the alzheimer-related protein tau. *Nature communications*, 8(1):275, 2017.
- [31] Sumie Yoshioka, Yukio Aso, Ken-ichi Izutsu, and S Kojima. Is stability prediction possible for protein drugs? denaturation kinetics of β -galactosidase in solution. *Pharmaceutical research*, 11:1721–1725, 1994.

- [32] Mireille Weijers, Peter A Barneveld, Martien A Cohen Stuart, and Ronald W Visschers. Heat-induced denaturation and aggregation of ovalbumin at neutral ph described by irreversible first-order kinetics. *Protein Science*, 12(12):2693–2703, 2003.
- [33] Nafisa Begam et al. Effects of temperature and ionic strength on the microscopic structure and dynamics of egg white gels. *The Journal of Chemical Physics*, 158(7), 2023.
- [34] Wei Wang and Christopher J Roberts. Non-arrhenius protein aggregation. *The AAPS journal*, 15: 840–851, 2013.
- [35] Pascal Blanpain-Avet et al. Predicting the distribution of whey protein fouling in a plate heat exchanger using the kinetic parameters of the thermal denaturation reaction of β -lactoglobulin and the bulk temperature profiles. *Journal of Dairy Science*, 99(12):9611–9630, 2016.

REVIEWERS' COMMENTS

Reviewer #1 (Remarks to the Author):

The manuscript is well-revised, with additional discussion of the relationship between XPCS results and viscosity and microstructure. Several other comments have also been appropriately addressed. It is considered publishable.

Reviewer #2 (Remarks to the Author):

The authors mostly answered my questions.

It is a bit concerning that the interpolation method makes such a big difference for the TTC images. I also do not understand the difference between no interpolation and nearest neighbour. Also for the no interpolation I don't see the rectangular pixels that should make up the image.

I suspect the author used matplotlib with interpolation none with a pdf output file which does interpolate the data. See for example https://matplotlib.org/2.0.0/examples/images_contours_and_fields/interpolation_none_vs_nearest.html

This is a minor thing not worth delaying the manuscript for, as the analysis is not done directly on the image, but it would be good if the authors displayed the data in a way that minimizes manipulations, which can be deceiving.

So I recommend the manuscript to be accepted, but hope the authors improve the way they render the TTC images.

Reviewer #3 (Remarks to the Author):

With the revisions, the authors have provided even stronger evidence for their conclusions. The quality of the scientific work (experiment design, data collection/analysis) is extremely high. The clarification to

figures and critical parts of the analysis improve readability and reproducibility. Moreover, the clarifications help to make the manuscript more approachable.

It is this reviewer's opinion that the revised work unquestionably meets the technical criteria for publication in Nature Communications. The data are comprehensive and sufficiently validated with literature, additional measurements (viscosity & EM), and the simultaneous structural and dynamical findings of this study are aligned.

I feel the main scientific importance of this work is the demonstration of using XPCS to gain insight to the structure and dynamics of biomolecular processes without damaging the sample with the X-ray probe or creating X-ray induced processes. As such, the authors have produced a high-quality phase diagram of a non-equilibrium, irreversible process. The biomolecular findings are also a match for Nature Communications.

I am therefore able to support publication of this work in Nature Communications.

Review-2: Response to comments of reviewer2 on the manuscript "Exploring non-equilibrium processes and spatio-temporal scaling laws in heated egg yolk using coherent X-rays"

We thank all reviewers for their thorough and critical reading of our manuscript, and their positive feedback on review-1. In the following, we provide a point-by-point response to the remaining comments from reviewer2. Our responses are highlighted in blue and the changes in the manuscript are highlighted in magenta.

REVIEWERS' COMMENTS

Reviewer 2 (Remarks to the Author): The authors mostly answered my questions.

It is a bit concerning that the interpolation method makes such a big difference for the TTC images. I also do not understand the difference between no interpolation and nearest neighbour. Also for the no interpolation I don't see the rectangular pixels that should make up the image. I suspect the author used matplotlib with interpolation none with a pdf output file which does interpolate the data. See for example

https://matplotlib.org/2.0.0/examples/images_contours_and_fields/interpolation_none_vs_nearest.html

This is a minor thing not worth delaying the manuscript for, as the analysis is not done directly on the image, but it would be good if the authors displayed the data in a way that minimizes manipulations, which can be deceiving.

So I recommend the manuscript to be accepted, but hope the authors improve the way they render the TTC images.

Authors' response: We thank the reviewer for pointing out this issue and also for this useful link. Indeed, we have used matplotlib to plot the TTC images. We agree with the reviewer that some kind of interpolation is applied when an image is saved as a vector graphic such as .pdf as mentioned in the link provided by the reviewer. Hence to avoid these issues we generate a .png output. In Fig. R1, we have compared two cases - "interpolation=none" and "interpolation=nearest". In both cases, a ".png" image is created as output file. Clearly, Fig. R1(a) and Fig. R1(b) look almost similar.

We have implemented the reviewer's suggestions in the revised manuscript, and we hope that the changes are satisfactory and that the reviewer recommends our work for publication.

Fig. R1: Fig. 4a of the manuscript generated using (a) "interpolation=none" applied, and (b) "interpolation=nearest" applied. In both cases, a ".png" image is created as output file.

Changes in the manuscript: Fig. 4a of the manuscript is replaced with Fig. R1(a).